# A Unified Wasserstein Distributional Robustness Framework for Adversarial Training

Tuan Anh Bui[1], Trung Le[1], Quan Hung Tran[2], He Zhao[1], and Dinh Phung[1, 3]

[1]Monash University    [2]Adobe Research    [3]VinAI Research

## Abstract

It is well-known that deep neural networks (DNNs) are susceptible to adversarial attacks, exposing a severe fragility of deep learning systems. As the result, adversarial training (AT) method, by incorporating adversarial examples during training, represents a natural and effective approach to strengthen the robustness of a DNN-based classifier. However, most AT-based methods, notably PGD-AT and TRADES, typically seek a pointwise adversary that generates the worst-case adversarial example by independently perturbing each data sample, as a way to "probe" the vulnerability of the classifier. Arguably, there are unexplored benefits in considering such adversarial effects from an entire distribution. To this end, this paper presents a unified framework that connects Wasserstein distributional robustness with current state-of-the-art AT methods. We introduce a new Wasserstein cost function and a new series of risk functions, with which we show that standard AT methods are special cases of their counterparts in our framework. This connection leads to an intuitive relaxation and generalization of existing AT methods and facilitates the development of a new family of distributional robustness AT-based algorithms. Extensive experiments show that our distributional robustness AT algorithms robustify further their standard AT counterparts in various settings.[1]

## 1 Introduction

Despite remarkable performances of DNN-based deep learning methods, even the state-of-the-art (SOTA) models are reported to be vulnerable to adversarial attacks (Biggio et al., 2013; Szegedy et al., 2014; Goodfellow et al., 2015; Madry et al., 2018; Athalye et al., 2018; Zhao et al., 2019b; 2021a), which is of significant concern given the large number of applications of deep learning in real-world scenarios. Usually, adversarial attacks are generated by adding small perturbations to benign data but to change the predictions of the target model. To enhance the robustness of DNNs, various adversarial defense methods have been developed, recently Pang et al. (2019); Dong et al. (2020); Zhang et al. (2020b); Bai et al. (2020). Among a number of adversarial defenses, Adversarial Training (AT) is one of the most effective and widely-used approaches (Goodfellow et al., 2015; Madry et al., 2018; Shafahi et al., 2019; Tramèr & Boneh, 2019; Zhang & Wang, 2019; Xie et al., 2020). In general, given a classifier, AT can be viewed as a robust optimization process (Ben-Tal et al., 2009) of seeking a pointwise adversary (Staib & Jegelka, 2017) that generates the worst-case adversarial example by independently perturbing each data sample.

Different from AT, Distributional Robustness (DR) (Delage & Ye, 2010; Duchi et al., 2021; Gao et al., 2017; Gao & Kleywegt, 2016; Rahimian & Mehrotra, 2019) looks for a worst-case distribution that generates adversarial examples from a known uncertainty set of distributions located in the ball centered around the data distribution. To measure the distance between distributions, different kinds of metrics have been considered in DR, such as $f$-divergence (Ben-Tal et al., 2013; Miyato et al., 2015; Namkoong & Duchi, 2016) and Wasserstein distance (Shafieezadeh-Abadeh et al., 2015; Blanchet et al., 2019; Kuhn et al., 2019), where the latter has shown advantages over others on efficiency and simplicity (Staib & Jegelka, 2017; Sinha et al., 2018). Therefore, adversary in DR does not look for the perturbation of a specific data sample, but moves the entire distribution around the data distribution, thus, is expected to have better generalization than AT on unseen data samples

---

[1]Our code is available at https://github.com/tuananhbui89/Unified-Distributional-Robustness

(Staib & Jegelka, 2017; Sinha et al., 2018). Conceptually and theoretically, DR can be viewed as a generalization and better alternative to AT and several attempts (Staib & Jegelka, 2017; Sinha et al., 2018) have shed light on connecting AT with DR. However, to the best of our knowledge, practical DR approaches that achieve comparable peformance with SOTA AT methods on adversarial robustness have not been developed yet.

To bridge this gap, we propose a unified framework that connects distributional robustness with various SOTA AT methods. Built on top of Wasserstein Distributional Robustness (WDR), we introduce a new cost function of the Wasserstein distances and propose a unified formulation of the risk function in WDR, with which, we can generalize and encompass SOTA AT methods in the DR setting, including PGD-AT (Madry et al., 2018), TRADES (Zhang et al., 2019), MART (Wang et al., 2019) and AWP (Wu et al., 2020). With better generalization capacity of distributional robustness, the resulted AT methods in our DR framework are shown to be able to achieve better adversarial robustness than their standard AT counterparts.

The contributions of this paper are in both theoretical and practical aspects, summarized as follows: **1)** Theoretically, we propose a general framework that bridges distributional robustness and standard robustness achieved by AT. The proposed framework encompasses the DR versions of the SOTA AT methods and we prove that these AT methods are special cases of their DR counterparts. **2)** Practically, motivated by our theoretical study, we develop a novel family of algorithms that generalize the AT methods in the standard robustness setting, which have better generalization capacity. **3)** Empirically, we conduct extensive experiments on benchmark datasets, which show that the proposed AT methods in the distributional robustness setting achieve better performance than standard AT methods.

## 2 PRELIMINARIES

### 2.1 DISTRIBUTIONAL ROBUSTNESS

Distributional Robustness (DR) is an emerging framework for learning and decision-making under uncertainty, which seeks the worst-case expected loss among a ball of distributions, containing all distributions that are close to the empirical distribution (Gao et al., 2017). Wasserstein DR has been one of the most widely-used variant of DR, which has rich applications in (semi)-supervised learning (Blanchet & Kang, 2020; Chen & Paschalidis, 2018; Yang, 2020), generative modeling (Huynh et al., 2021; Dam et al., 2019), transfer learning and domain adaptation (Lee & Raginsky, 2018; Duchi et al., 2019; Zhao et al., 2019a; Nguyen et al., 2021a;b; Le et al., 2021b;a), topic modeling (Zhao et al., 2021b), and reinforcement learning (Abdullah et al., 2019; Smirnova et al., 2019; Derman & Mannor, 2020). For more comprehensive review, please refer to the surveys of Kuhn et al. (2019); Rahimian & Mehrotra (2019). Here we consider a generic Polish space $S$ endowed with a distribution $\mathbb{P}$. Let $f : S \to \mathbb{R}$ be a real-valued (risk) function and $c : S \times S \to \mathbb{R}_+$ be a cost function. Distributional robustness setting aims to find the distribution $\mathbb{Q}$ in the vicinity of $\mathbb{P}$ and maximizes the risk in the $\mathbb{E}$ form (Sinha et al., 2018; Blanchet & Murthy, 2019):

$$\sup_{\mathbb{Q}:\mathcal{W}_c(\mathbb{P},\mathbb{Q})<\epsilon} \mathbb{E}_{\mathbb{Q}}\left[f\left(z\right)\right], \tag{1}$$

where $\epsilon > 0$ and $\mathcal{W}_c$ denotes the optimal transport (OT) cost, or a Wasserstein distance if $c$ is a metric, defined as:

$$\mathcal{W}_c\left(\mathbb{P}, \mathbb{Q}\right) := \inf_{\gamma \in \Gamma(\mathbb{P},\mathbb{Q})} \int c d\gamma, \tag{2}$$

where $\Gamma\left(\mathbb{P}, \mathbb{Q}\right)$ is the set of couplings whose marginals are $\mathbb{P}$ and $\mathbb{Q}$. With the assumption that $f \in L^1\left(\mathbb{P}\right)$ is upper semi-continuous and the cost $c$ is a non-negative lower semi-continuous satisfying $c(z, z') = 0$ iff $z = z'$, Sinha et al. (2018); Blanchet & Murthy (2019) show that the *dual* form for Eq. (1) is:

$$\inf_{\lambda \geq 0} \left\{\lambda \epsilon + \mathbb{E}_{z \sim \mathbb{P}}\left[\sup_{z'} \left\{f\left(z'\right) - \lambda c\left(z', z\right)\right\}\right]\right\}. \tag{3}$$

Sinha et al. (2018) further employs a Lagrangian for Wasserstein-based uncertainty sets to arrive at a relaxed version with $\lambda \geq 0$:

$$\sup_{\mathbb{Q}} \left\{\mathbb{E}_{\mathbb{Q}}\left[f\left(z\right)\right] - \lambda \mathcal{W}_c\left(\mathbb{P}, \mathbb{Q}\right)\right\} = \mathbb{E}_{z \sim \mathbb{P}}\left[\sup_{z'} \left\{f\left(z'\right) - \lambda c\left(z', z\right)\right\}\right]. \tag{4}$$

## 2.2 Adversarial Robustness with Adversarial Training

In this paper, we are interested in image classification tasks and focus on the adversaries that add small perturbations to the pixels of an image to generate attacks based on gradients, which are the most popular and effective. FGSM (Goodfellow et al., 2015) and PGD (Madry et al., 2018) are the most representative gradient-based attacks and PGD is the most widely-used one, due to its effectiveness and simplicity. Now we consider a classification problem on the space $S = \mathcal{X} \times \mathcal{Y}$ where $\mathcal{X}$ is the data space, $\mathcal{Y}$ is the label space. We would like to learn a classifier that predicts the label of a datum well $h_\theta : \mathcal{X} \to \mathcal{Y}$. Learning of the classifier can be done by minimising its loss: $\ell\left(h_\theta\left(x\right), y\right)$, which can typically be the the cross-entropy loss. In addition to predicting well on benign data, an adversarial defense aims to make the classifier robust against adversarial examples. As the most successful approach, adversarial training is a straightforward method that creates and then incorporates adversarial examples into the training process. With this general idea, different AT methods vary in the way of picking which adversarial examples one should train on. Here we list three widely-used AT methods.

**PGD-AT** (Madry et al., 2018) seeks the *most violating* examples to improve model robustness:

$$\inf_\theta \mathbb{E}_\mathbb{P} \left[ \beta \sup_{x' \in B_\epsilon(x)} CE\left(h_\theta\left(x'\right), y\right) + CE\left(h_\theta\left(x\right), y\right) \right], \tag{5}$$

where $B_\epsilon\left(x\right) = \{x' : c_\mathcal{X}\left(x, x'\right) \leq \epsilon\}$, $\beta > 0$ is the trade-off parameter and cross-entropy loss CE.

**TRADES** (Zhang et al., 2019) seeks the *most divergent* examples to improve model robustness:

$$\inf_\theta \mathbb{E}_\mathbb{P} \left[ \beta \sup_{x'} D_{KL}\left(h_\theta\left(x'\right), h_\theta\left(x\right)\right) + CE\left(h_\theta\left(x\right), y\right) \right], \tag{6}$$

where $x' \in B_\epsilon\left(x\right)$ and $D_{KL}$ is the usual Kullback-Leibler (KL) divergence.

**MART** (Wang et al., 2019) takes into account prediction confidence:

$$\inf_\theta \mathbb{E}_\mathbb{P} \left[ \beta \left(1 - \left[h_\theta\left(x\right)\right]_y\right) \sup_{x' \in B_\epsilon(x)} D_{KL}\left(h_\theta\left(x'\right), h_\theta\left(x\right)\right) + BCE\left(h_\theta\left(x\right), y\right) \right], \tag{7}$$

where $BCE\left(h_\theta\left(x\right), y\right)$ is defined as: $-\log\left(\left[h_\theta\left(x\right)\right]_y\right) - \log\left(1 - \max_{k \neq y}\left[h_\theta\left(x\right)\right]_k\right)$.

## 2.3 Connecting Distributional Robustness to Adversarial Training

To bridge distributional and adversarial robustness, Sinha et al. (2018) proposes an AT method, named Wasserstein Risk Minimization (WRM), which generalizes PGD-AT through the principled lens of distributionally robust optimization. For smooth loss functions, WRM enjoys convergence guarantees similar to non-robust approaches while certifying performance even for the worst-case population loss. Specifically, assume that $\mathbb{P}$ is a joint distribution that generates a pair $z = (x, y)$ where $x \in \mathcal{X}$ and $y \in \mathcal{Y}$. The cost function is defined as: $c\left(z, z'\right) = c_\mathcal{X}\left(x, x'\right) + \infty \times \mathbf{1}\left\{y \neq y'\right\}$ where $z' = (x', y')$, $c_\mathcal{X} : \mathcal{X} \times \mathcal{X} \to \mathbb{R}_+$ is a cost function on $\mathcal{X}$, and $\mathbf{1}\left\{\cdot\right\}$ is the indicator function. One can define the risk function $f$ as the loss of the classifier, i.e., $f\left(z\right) := \ell\left(h_\theta\left(x\right), y\right)$. Together with Eq. (1), attaining a robust classifier is to solve the following min-max problem:

$$\inf_\theta \sup_{\mathbb{Q}:\mathcal{W}_c(\mathbb{P},\mathbb{Q})<\epsilon} \mathbb{E}_\mathbb{Q}\left[\ell\left(h_\theta\left(x\right), y\right)\right]. \tag{8}$$

The above equation shows the generalisation of WRM to PGD-AT. With Eq. (3) and Eq. (4), one can arrive at Eq. (9) as below where $\lambda \geq 0$ is a trade-off parameter:

$$\inf_\theta \mathbb{E}_\mathbb{P} \left[ \sup_{x'} \left\{\ell\left(h_\theta\left(x'\right), y\right) - \lambda c_\mathcal{X}\left(x', x\right)\right\} \right]. \tag{9}$$

## 3 Proposed Unified Distribution Robustness Framework

Although WRM (Sinha et al., 2018) sheds light on connecting distributional robustness with adversarial training, its framework and formulation is limited to PGD-AT, which cannot encompass more

advanced AT methods including TRADES and MART. In this paper, we propose a unified formulation for distributional robustness, which is a more general framework connecting state-of-the-art AT and existing distributional robustness approaches where they become special cases.

Let $\mathbb{P}^d$ be the data distribution that generates instance $x \sim \mathbb{P}^d$ and $\mathbb{P}^l_{\cdot|x}$ the conditional to generate label $y \sim \mathbb{P}^l_{\cdot|x}$ given $x$ where $x \in \mathcal{X}, y \in \mathcal{Y}$. For our purpose, we consider the space $S = \mathcal{X} \times \mathcal{X} \times \mathcal{Y}$ and a joint distribution $\mathbb{P}_\triangle$ on $S$ consisting of samples $(x, x, y)$ where $x \sim \mathbb{P}^d$ and $y \sim \mathbb{P}^l_{\cdot|x}$. Now consider a distribution $\mathbb{Q}$ on $S$ such that $\mathcal{W}_c (\mathbb{Q}, \mathbb{P}_\triangle) < \epsilon$. A draw $z \sim \mathbb{P}_\triangle$ will take the form $z = (x, x, y)$ whereas $z' \sim \mathbb{Q}$ will be $z' = (x', x'', y')$. We propose cost function $c(z, z')$ defined as:

$$c(z, z') = c_\mathcal{X}(x, x') + \infty \times c_\mathcal{X}(x, x'') + \infty \times \mathbf{1}\{y \neq y'\}, \tag{10}$$

where we note that this cost function is non negative, satisfies $c(z, z) = 0$ and lower semi-continuous, i.e., $\lim_{z' \to z_0} \inf c(z, z') \geq c(z, z_0)$.

With our new setting, it is useful to understand the "vicinity"of $\mathbb{P}_\triangle$ via the distribution OT-ball condition $\mathcal{W}_c(\mathbb{Q}, \mathbb{P}_\triangle) < \epsilon$. Since there exists a transport plan $\gamma \in \Gamma(\mathbb{P}_\triangle, \mathbb{Q})$ s.t. $\int c d\gamma < \epsilon$ and $c(z, z')$ is finite a.s. $\gamma$, this implies that if $(z, z') \sim \gamma$, then first, it is easy to see that $x'' = x$ and $y' = y$, and second, $x'$ tends to be close to $x$. To see why the later is the case, since $\mathbb{P}^d$ is a marginal of $\mathbb{P}_\triangle$ on the first $x$ in $(x, x, y)$, therefore if $\mathbb{Q}^d$ is the marginal of $\mathbb{Q}$ on $x'$ in $(x', x'', y')$ then $\mathcal{W}_d(\mathbb{Q}^d, \mathbb{P}^d) \leq \mathcal{W}_d(\mathbb{Q}, \mathbb{P}_\triangle) < \epsilon$, which explains the closeness between of $x$ and $x'$.

Given $z' = (x', x'', y') \sim \mathbb{Q}$ where $\mathcal{W}_c(\mathbb{Q}, \mathbb{P}_\triangle) < \epsilon$, we define a unified risk function $g_\theta(z')$ w.r.t a classifier $h_\theta$ that encompasses the unified distributional robustness (UDR) version for PGD-AT, TRADES, and MART (cf Section 2.2):

- *UDR-PGD*: $g_\theta(z') := CE(h_\theta(x''), y') + \beta CE(h_\theta(x'), y')$.
- *UDR-TRADES*: $g_\theta(z') := CE(h_\theta(x''), y') + \beta D_{KL}(h_\theta(x'), h_\theta(x''))$.
- *UDR-MART*: $g_\theta(z') := BCE(h_\theta(x''), y') + \beta(1 - [h_\theta(x'')]_y)D_{KL}(h_\theta(x'), h_\theta(x''))$.[2]

Now we derive the primal and dual objectives for the proposed UDR framework. With the UDR risk function $g_\theta(z')$ defined previously, following Eq. (1) and Eq. (3), the primal (left) and dual (right) forms of our UDR objective are:

$$\inf_\theta \sup_{\mathbb{Q}:\mathcal{W}_c(\mathbb{Q},\mathbb{P}_\triangle)<\epsilon} \mathbb{E}_\mathbb{Q}[g_\theta(z')] = \inf_\theta \inf_{\lambda \geq 0} \left( \lambda\epsilon + \mathbb{E}_{\mathbb{P}_\triangle}\left[ \sup_{z'}\{g_\theta(z') - \lambda c(z', z)\} \right] \right). \tag{11}$$

With the cost function $c$ defined in Eq. (10), the dual form in (11) can be rewritten as:

$$\inf_{\theta,\lambda \geq 0} \left( \lambda\epsilon + \mathbb{E}_{\mathbb{P}_\triangle}\left[ \sup_{x',x''=x,y'=y}\{g_\theta(z') - \lambda c_\mathcal{X}(x', x)\} \right] \right) =$$
$$\inf_{\theta,\lambda \geq 0} \left( \lambda\epsilon + \mathbb{E}_\mathbb{P}\left[ \sup_{x'}\{g_\theta(x', x, y) - \lambda c_\mathcal{X}(x', x)\} \right] \right) \tag{12}$$

where we note that $\mathbb{P}$ is a distribution over pairs $(x, y)$ for which $x \sim \mathbb{P}^d$ and $y \sim \mathbb{P}^l_{\cdot|x}$. The min-max problem in Eq. (12) encompasses the PGD-AT, TRADES, and MART distributional robustness counterparts on the choice of the function $g_\theta(x', x, y)$ by simply choosing an appropriate $g_\theta(x', x, y)$ as shown in Section 2.3.

In what follows, we prove that standard PGD-AT, TRADES, and MART presented in Section 2 are specific cases of their UDR counterparts by specifying corresponding cost functions. Given a cost function $c_\mathcal{X}$ (e.g., $L_1, L_2$, and $L_\infty$), we define a new cost function $\tilde{c}_\mathcal{X}$ as:

$$\tilde{c}_\mathcal{X}(x, x') = \begin{cases} c_\mathcal{X}(x, x') & \text{if } c_\mathcal{X}(x, x') \leq \epsilon \\ \infty & \text{otherwise.} \end{cases} \tag{13}$$

The cost function $\tilde{c}_\mathcal{X}$ is lower semi-continuous. By defining the ball $B_\epsilon(x) := \{x' : c_\mathcal{X}(x, x') \leq \epsilon\} = \{x' : \tilde{c}_\mathcal{X}(x, x') \leq \epsilon\}$, we achieve the following theorem on the relation between distributional and standard robustness.

---

[2]To encompass MART with our framework, we assume a classifier is adversarially trained by Eq. (7) with adversarial examples generated by $\sup_{x' \in B_\epsilon(x)} D_{KL}(h_\theta(x'), h_\theta(x)) + BCE(h_\theta(x), y)$. This is slightly different from the original MART, where the adversarial examples are generated by $\sup_{x' \in B_\epsilon(x)} CE(h_\theta(x'), y)$.

**Theorem 1.** *With the cost function $\tilde{c}_x$ defined as above, the optimization problem:*

$$\inf_{\theta, \lambda \geq 0} \left( \lambda \epsilon + \mathbb{E}_{\mathbb{P}} \left[ \sup_{x'} \{ g_\theta \left( x', x, y \right) - \lambda \tilde{c}_{\mathcal{X}} \left( x', x \right) \} \right] \right) \tag{14}$$

*is equivalent to the optimization problem:*

$$\inf_{\theta} \mathbb{E}_{\mathbb{P}} \left[ \sup_{x' \in B_\epsilon(x)} g_\theta \left( x', x, y \right) \right]. \tag{15}$$

*Proof.* See Appendix A for the proof. □

**Theoretical contribution and comparison to previous work.** Theorem 1 says that the standard PGD-AT, TRADES, and MART are special cases of their UDR counterparts, which indicates that our UDR versions of AT have a richer expressiveness capacity than the standard ones. Different from WRM (Sinha et al., 2018) , our proposed framework is developed based on theoretical foundation of (Blanchet & Murthy, 2019). It is worth noting that the theoretical development is *not trivial* because theory developed in Blanchet & Murthy (2019) is only valid for a bounded cost function, while the cost function $\tilde{c}$ is unbounded. More specifically, the transformation from primal to dual forms in Eq. (11) requires the cost function $c$ to be bounded. In Theorem 2 in Appendix A, we prove this primal-dual form transformation for the unbounded cost function $\tilde{c}_{\mathcal{X}}$, which is certainly not trivial.

Moreover, our UDR is *fundamentally distinctive* from WRM in its ability to adapt and learn $\lambda$, while this is a hyper-parameter in WRM. As a result of a fixed $\lambda$, WRM is fundamentally same as PGD in the sense that these methods can only utilize local information of relevant benign examples when crafting adversarial examples. In contrast, our UDR can leverage both local and global information of multiple benign examples when crafting adversarial examples due to the fact that $\lambda$ is adaptable and captures the global information when solving the outer minimization in (14). Further explanation can be found in Appendix B.

## 4    LEARNING ROBUST MODELS WITH UDR

In this section we introduce the details of how to learn robust models with UDR. To do this, we first discuss the induced cost function $\tilde{c}_{\mathcal{X}}$ defined as in Eq (13), which assists us in understanding the connection between distributional and standard robustness approaches. We note that $\tilde{c}_{\mathcal{X}}$ is non-differential outside the perturbation ball (i.e., $c_{\mathcal{X}}(x', x) \geq \epsilon$). To circumvent this, we introduce a smoothed version $\hat{c}_{\mathcal{X}}$ to approximate $\tilde{c}_{\mathcal{X}}$ as follows:

$$\hat{c}_{\mathcal{X}} \left( x, x' \right) := \mathbf{1} \left\{ c_{\mathcal{X}}(x, x') < \epsilon \right\} c_{\mathcal{X}}(x, x') + \mathbf{1} \left\{ c_{\mathcal{X}}(x, x') \geq \epsilon \right\} \left( \epsilon + \frac{c_{\mathcal{X}}(x, x') - \epsilon}{\tau} \right), \tag{16}$$

where $\tau > 0$ is the temperature to control the growing rate of the cost function when $x'$ goes out of the perturbation ball. It is obvious that $\hat{c}_{\mathcal{X}} \left( x, x' \right)$ is continuous and approaches $\tilde{c}_{\mathcal{X}} \left( x, x' \right)$ when $\tau \to 0$. Using the smoothed function $\hat{c}_{\mathcal{X}} \left( x, x' \right)$ from Eq. (16), the final object of our UDR becomes:

$$\inf_{\theta, \lambda \geq 0} \left( \lambda \epsilon + \mathbb{E}_{\mathbb{P}} \left[ \sup_{x'} \{ g_\theta \left( x', x, y \right) - \lambda \hat{c}_{\mathcal{X}} \left( x', x \right) \} \right] \right). \tag{17}$$

With this final objective, our training strategy involves three iterative steps at each iteration w.r.t. a batch of data examples, which are shown in Algorithm 1.

**1. Craft adversarial examples w.r.t. the current model and the parameter $\lambda$.** Given the current model $\theta$ and the parameter $\lambda$, we find the adversarial examples by solving:

$$x^a = \operatorname{argmax}_{x'} \{ g_\theta(x', x, y) - \lambda \hat{c}_{\mathcal{X}} \left( x', x \right) \}, \tag{18}$$

where different methods (i.e., UDR-PGD, UDR-TRADES, etc.) specifies $g_\theta(x', x, y)$ differently.

Similar to other AT methods like PGD-AT, we employ iterative gradient ascent update steps to optimise to find $x^a$. Specifically, we start from a random example inside the ball $B_\epsilon$ and update in $k$ steps with the step size $\eta > 0$. Since the magnitude of the gradient $\nabla_{x'} g_\theta(x', x, y)$ is significantly smaller than that of $\nabla_{x'} \hat{c}_{\mathcal{X}}(x', x)$, we use sign $(\nabla_{x'} \hat{c}_{\mathcal{X}}(x', x))$ in the update formula rather than $\nabla_{x'} \hat{c}_{\mathcal{X}}(x', x)$. These steps are shown in 2(a) to 2(c) of Algorithm 1.

An important difference from ours to other AT methods is that at each update step, we do not apply any explicit projecting operations onto the ball $B_\epsilon$. Indeed, the parameter $\lambda$ controls how distant $x^a$ to its benign counterpart $x$. Thus, this can be viewed as implicitly projecting onto a soft ball governed by the magnitude of the parameter $\lambda$ and the temperature $\tau$. Specifically, when $\lambda$ becomes higher, the crafted adversarial examples $x^a$ stay closer to their benign counterparts $x$ and vice versa. When $\tau$ is set closer to 0, the smoothed cost function $\hat{c}_\mathcal{X}$ approximates the cost function $\tilde{c}_\mathcal{X}$ more tightly. Thus, our soft-ball projection is more identical to the hard ball projection as in projected gradient ascent.

---

**Algorithm 1** The pseudocode of our proposed method.

**Input**: training set $\mathcal{D}$, number of iterations $T$, batch size $N$, adversary parameters $\{k, \epsilon, \eta\}$

**for** $t = 1$ to $T$ **do**

1. **Sample** mini-batch $\{x_i, y_i\}_{i=1}^N \sim \mathcal{D}$
2. **Find** adversarial examples $\{x_i^a\}_{i=1}^N$ using Eq. (18)
   (a) **Initialize** randomly: $x_i^0 = x_i + noise$ where $noise \sim \mathcal{U}(-\epsilon, \epsilon)$
   (b) **for** $n = 1$ to $k$ **do**
       i. $x_i^{inter} = x_i^n + \eta \text{sign}\left(\nabla_x g_\theta(x_i^n, x_i, y_i)\right)$
       ii. $x_i^{n+1} = x_i^{inter} - \eta \lambda \nabla_x \hat{c}(x_i^{inter}, x_i)$
   (c) **Clip** to valid range: $x_i^a = clip(x_i^k, 0, 1)$
3. **Update** parameter $\lambda$ using Eq. (19)
4. **Update** model parameter $\theta$ using Eq. (20)

**Output:** model parameter $\theta$

---

**2. Update the parameter $\lambda$.** Given current model $\theta$, we craft a batch of adversarial examples $\{x_i^a\}_{i=1}^N$ corresponding to the benign examples $\{x_i\}_{i=1}^N$ crafted as above. Inspired by the Danskin's theorem , we update $\lambda$ as follows:

$$\lambda_n = \lambda - \eta_\lambda \left(\epsilon - \frac{1}{N} \sum_{i=1}^N \hat{c}_\mathcal{X}(x_i^a, x_i)\right), \tag{19}$$

where $\eta_\lambda > 0$ is a learning rate and $\lambda_n$ represents the new value of $\lambda$.

The proposed update of $\lambda$ is intuitive: *if the adversarial examples stay close to their benign examples, i.e., $\sum_{i=1}^N \hat{c}_\mathcal{X}(x_i^a, x_i) < \epsilon$, $\lambda$ decreases to make them more distant to the benign examples and vice versa.* Therefore the adversarial examples are crafted more diversely, which can further strengthen the robustness of the model.

**3. Update the model parameter $\theta$.** Given the set of adversarial examples $\{x_i^a\}_{i=1}^N$ crafted as above and their benign examples $\{x_i\}_{i=1}^N$ with the labels $\{y_i\}_{i=1}^N$, we update the model parameter $\theta$ to minimize $\mathbb{E}_\mathbb{P}\left[\nabla g_\theta(x^a, x, y)\right]$ using the current batches of adversarial and benign examples:

$$\theta_n = \theta - \frac{\eta_\theta}{N} \sum_{i=1}^N \nabla_\theta g_\theta(x_i^a, x_i, y_i), \tag{20}$$

where $\eta_\theta > 0$ is a learning rate and $\theta_n$ specifies the new model parameter.

## 5 EXPERIMENTS

We use MNIST (LeCun et al., 1998), CIFAR10 and CIFAR100 (Krizhevsky et al., 2009) as the benchmark datasets in our experiment. The inputs were normalized to $[0, 1]$. We apply padding 4 pixels at all borders before random cropping and random horizontal flips as used in Zhang et al. (2019). We use both standard CNN architecture (Carlini & Wagner, 2017) and ResNet architecture (He et al., 2016) in our experiment. The architecture and training setting are provided in Appendix D.

We compare our UDR with the SOTA AT methods, i.e., **PGD-AT** (Madry et al., 2018), **TRADES** (Zhang et al., 2019) and **MART** (Wang et al., 2019). Because TRADES and MART performances are strongly dependent on the trade-off ratio (i.e., $\beta$ in Eq. (6) and (7)) between natural loss and robust loss, we use the original setting in their papers (CIFAR10/CIFAR100: $\beta = 6$ for TRADES/UDR-TRADES, $\beta = 5$ for MART/UDR-MART; MNIST: $\beta = 1$ for all the methods). We also tried with the distributional robustness method WRM (Sinha et al., 2018). However, WRM did not seem to obtain reasonable performance in our experiments. Its results can be found in Appendix F. For all the AT methods, we use $\{k = 40, \epsilon = 0.3, \eta = 0.01\}$ for the MNIST dataset,

Table 1: Comparisons of natural classification accuracy (Nat) and adversarial accuracies against different attacks. Best scores are highlighted in boldface.

| | MNIST | | | | CIFAR10 | | | | CIFAR100 | | | |
|---|---|---|---|---|---|---|---|---|---|---|---|---|
| | Nat | PGD | AA | B&B | Nat | PGD | AA | B&B | Nat | PGD | AA | B&B |
| PGD-AT | 99.4 | 94.0 | 88.9 | 91.3 | **86.4** | 46.0 | 42.5 | 44.2 | 72.4 | 41.7 | 39.3 | 39.6 |
| UDR-PGD | **99.5** | **94.3** | **90.0** | **91.4** | **86.4** | **48.9** | **44.8** | **46.0** | **73.5** | **45.1** | **41.9** | **42.3** |
| TRADES | **99.4** | 95.1 | 90.9 | 92.2 | 80.8 | 51.9 | 49.1 | 50.2 | 68.1 | 49.7 | 46.7 | 47.2 |
| UDR-TRADES | **99.4** | **96.9** | **92.2** | **95.2** | **84.4** | **53.6** | **49.9** | **51.0** | **69.6** | **49.9** | **47.8** | **48.7** |
| MART | **99.3** | 94.7 | 90.6 | 92.9 | **81.9** | 53.3 | 48.2 | 49.3 | **68.1** | 49.8 | 44.8 | 45.4 |
| UDR-MART | **99.3** | **96.0** | **92.3** | **94.4** | 80.1 | **54.1** | **49.1** | **50.4** | 67.5 | **52.0** | **48.5** | **48.6** |

Table 2: Robustness evaluation under different PGD attack strengths $\epsilon$. *Avg* represents for the average improvement of our DR methods over their counterparts.

| MNIST | | | | | | | |
|---|---|---|---|---|---|---|---|
| $\epsilon$ | 0.3 | 0.325 | 0.35 | 0.375 | 0.4 | 0.425 | Avg |
| PGD-AT | 94.0 | 67.8 | 21.1 | 6.8 | 2.3 | 1.2 | - |
| UDR-PGD | **94.3** | **92.9** | **90.1** | **79.2** | **22.3** | **3.8** | 31.57 |
| TRADES | 95.5 | 85.2 | 34.4 | 5.8 | 0.6 | 0.1 | - |
| UDR-TRADES | **96.9** | **96.9** | **95.8** | **95.1** | **94.5** | **88.5** | 57.68 |
| MART | 94.7 | 66.1 | 9.4 | 0.9 | 0.2 | 0.1 | - |
| UDR-MART | **96.0** | **95.0** | **94.1** | **92.8** | **88.8** | **37.7** | 55.5 |

| CIFAR10 | | | | | | | |
|---|---|---|---|---|---|---|---|
| $\epsilon$ | $\frac{8}{255}$ | $\frac{10}{255}$ | $\frac{12}{255}$ | $\frac{14}{255}$ | $\frac{16}{255}$ | $\frac{20}{255}$ | Avg |
| PGD-AT | 46.0 | 33.7 | 23.7 | 15.2 | 9.5 | 3.6 | - |
| UDR-PGD | **48.9** | **36.4** | **26.3** | **18.5** | **13.0** | **7.1** | 3.08 |
| TRADES | 51.9 | 42.5 | 33.7 | 25.7 | 18.9 | 9.1 | - |
| UDR-TRADES | **53.6** | **43.6** | **35.2** | **27.5** | **20.7** | **10.9** | 1.62 |
| MART | 53.3 | 43.2 | 34.1 | 25.5 | 18.4 | 9.0 | - |
| UDR-MART | **54.1** | **46.0** | **37.3** | **29.7** | **22.9** | **12.2** | 3.12 |

| CIFAR100 | | | | | | | |
|---|---|---|---|---|---|---|---|
| $\epsilon$ | $\frac{10}{1000}$ | $\frac{12.5}{1000}$ | $\frac{15}{1000}$ | $\frac{17.5}{1000}$ | $\frac{20}{1000}$ | $\frac{25}{1000}$ | Avg |
| PGD-AT | 41.7 | 34.5 | 27.8 | 22.6 | 18.2 | 11.7 | - |
| UDR-PGD | **45.1** | **38.3** | **31.9** | **26.2** | **21.4** | **14.2** | 3.43 |
| TRADES | 49.7 | 44.3 | 39.9 | 35.2 | 31.2 | 23.5 | - |
| UDR-TRADES | **49.9** | **44.8** | **40.3** | **35.7** | **31.7** | **24.2** | 0.47 |
| MART | 49.8 | 45.3 | 41.0 | 36.6 | 32.4 | 25.1 | - |
| UDR-MART | **52.0** | **47.8** | **44.1** | **40.2** | **36.2** | **29.4** | 3.25 |

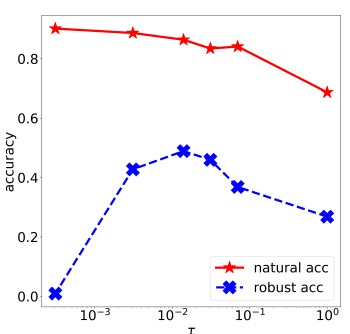

(a) Natural/robust accuracy trade-off

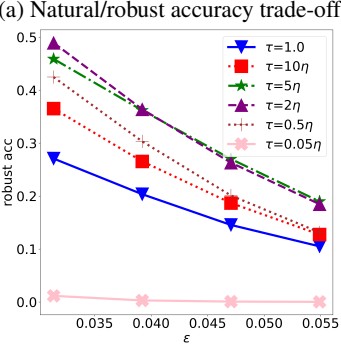

(b) Robustness in correlation with $\tau$

Figure 1: Further analysis on parameter sensitivity.

$\{k = 10, \epsilon = 8/255, \eta = 2/255\}$ for the CIFAR10 dataset and $\{k = 10, \epsilon = 0.01, \eta = 0.001\}$ for the CIFAR100 dataset, where $k$ is number of iteration, $\epsilon$ is the distortion bound and $\eta$ is the step size of the adversaries.

We use different SOTA attacks to evaluate the defense methods including: **1) PGD attack** (Madry et al., 2018) which is one of the most widely-used gradient based attacks. For PGD, we set $k = 200$ and $\epsilon = 0.3, \eta = 0.01$ for MNIST, $\epsilon = 8/255, \eta = 2/255$ for CIFAR10, and $\epsilon = 0.01, \eta = 0.001$ for CIFAR100, which are the standard settings. **2) B&B attack** (Brendel et al., 2019) which is a decision based attack. Following Tramer et al. (2020), we initialized with the PGD attack with $k = 20$ and corresponding $\{\epsilon, \eta\}$ then apply B&B attack with 200 steps. **3) Auto-Attack (AA)** (Croce & Hein, 2020b) which is an ensemble methods of four different attacks. We use $\epsilon = 0.3, 8/255, 0.01$, for MNIST, CIFAR10, and CIFAR100, respectively. The distortion metric we use in our experiments is $l_\infty$ for all measures. We use the full test set for PGD and 1000 test samples for the other attacks.

Table 3: Adversarial accuracy in the blackbox settings. *Avg* represents for the average improvement of our DR methods over their counterparts.

| Source / Target | PGD-AT | UDR-P | TRADES | UDR-T | MART | UDR-M | Avg |
|---|---|---|---|---|---|---|---|
| PGD-AT | - | - | 61.6 | 61.6 | 61.7 | 62.4 | - |
| UDR-PGD | - | - | **63.6** | **63.4** | **64.0** | **64.1** | 2.0 |
| TRADES | 61.2 | 61.3 | - | - | 58.9 | 59.8 | - |
| UDR-TRADES | **62.7** | **62.8** | - | - | **61.1** | **61.6** | 1.8 |
| MART | 61.4 | 61.4 | 58.9 | 59.5 | - | - | - |
| UDR-MART | **62.3** | **62.1** | **60.1** | **60.5** | - | - | 1.0 |

## 5.1 MAIN RESULTS

**Whitebox Attacks with fixed $\epsilon$.** First, we compare the natural and robust accuracy of the AT methods and their counterparts under our UDR framework, against several SOTA attacks. Note that in this experiment, the attacks are with their standard settings. The result of this experiment is shown in Table 1. It can be observed that for all the AT methods, our UDR versions are able to boost the model robustness significantly against all the strong attack methods in comparison on all the three datasets. These improvements clearly show that our UDR empowered AT methods achieve the SOTA adversarial robustness performance. Specifically, our UDR-PGD's improvement over PGD on both CIFAR10 and CIFAR100 is over 3% against all the attacks. Similarly, our UDR-MART also improves over MART with a 3% gap on CIFAR100.

**Whitebox Attacks with varied $\epsilon$.** Recall that UDR is designed to have better generalization capacity than standard adversarial robustness. In this experiment, we exam the generalization capacity by attacking the AT methods (including our UDR variants) with PGD with varied attack strength $\epsilon$ while keeping other parameters of PGD attack the same. This is a highly practical scenario where attackers may use various attack strengths that are different from that the model is trained with. The results of this experiment are shown in Table 2. We have the following remarks of the results: **1)** All AT methods perform reasonably well (our UDR variants are better than their counterparts) when PGD attacks with the same $\epsilon$ that these methods are trained on. This is shown in the first column on all the datasets, whose results are in line with these in Table 1. **2)** With increased $\epsilon$, the performance of all the AT methods deteriorates, which is natural. However, the advantage of our UDR methods over their counterparts becomes more and more significant. For example, when $\epsilon = 0.375$, all of our UDR methods can achieve at least 80% robust accuracy on MNIST, while others can barely defend. This clearly demonstrates the benefit of our UDR framework on generalization capacity.

**Blackbox Attacks.** To further exam the generalization of the UDR framework, we conduct the experiment with the blackbox setting via transferred attacks. Specifically, we use PGD to generate adversarial examples according to the model trained with a specific AT method, i.e., the *source* method. Next, we use the generated adversarial examples to attack another AT method, i.e., the *target* method. This is to see whether an AT method can defend against attacks generated from other models. We report the results in Table 3. It can be seen that with better generalization capacity, our UDR methods also outperform their standard counterparts with a margin of 2% in the blackbox setting.

**Results with WideResNet architecture.** We would like to provide further experimental results on the CIFAR10 dataset with WideResNet (WRN-34-10) as shown in Table 4. It can be seen that our distributional frameworks consistently outperform their standard AT counterparts in both metrics. More specifically, our improvement over PGD-AT against Auto-Attack is around 0.8%, while that for TRADES is 0.5%. To make a more concrete conclusion, we deploy our framework on a recent SOTA standard AT which is AWP-AT Wu et al. (2020). The result shows that our distributional

Table 4: Robustness evaluation against Auto-Attack and PGD ($k = 100$) with WRN-34-10 on the full test set of CIFAR10 dataset. (*) Omit the cross-entropy loss of natural images. Detail can be found in Appendix D.

| | Nat | PGD | AA | C&W |
|---|---|---|---|---|
| PGD-AT* | 84.93 | 55.04 | 52.12 | 40.85 |
| UDR-PGD* | 84.60 | 55.71 | 52.98 | 47.31 |
| TRADES | 85.70 | 56.97 | 53.82 | 47.65 |
| UDR-TRADES | 84.93 | 57.35 | 54.45 | 49.14 |
| AWP-AT | 85.57 | 57.78 | 53.91 | 49.91 |
| UDR-AWP-AT | 85.51 | 58.65 | 54.40 | 54.44 |
| Zhang et al. (2020a) | 84.52 | - | 53.51 | - |
| Huang et al. (2020) | 83.48 | - | 53.34 | - |
| Zhang et al. (2019) | 84.92 | - | 53.08 | - |
| Cui et al. (2021) | 88.22 | - | 52.86 | - |

Table 5: Average norm $L_1$ and $L_\infty$ of the perturbation $\delta = |x^a - x|_p$

|  | $L_1$ | $L_\infty$ | $p(\delta \leq 0.9\epsilon)$ | $p(\delta \leq \epsilon)$ | $p(\delta \leq 1.1\epsilon)$ |
|---|---|---|---|---|---|
| PGD | 0.0270 | 0.031 | 19.7% | 100% | 100% |
| UDR-PGD at epoch 0th | 0.0278 | 0.031 | 18.9% | 100% | 100% |
| UDR-PGD at epoch 200th | 0.0301 | 0.034 | 19.5% | 22.1% | 100% |

Table 6: Comparison to PGD-AT with different perturbation limitations.

|  | $\frac{8}{255}$ | $\frac{10}{255}$ | $\frac{12}{255}$ | $\frac{14}{255}$ | $\frac{16}{255}$ | $\frac{20}{255}$ | Avg |
|---|---|---|---|---|---|---|---|
| PGD-AT at $\epsilon = 0.031$ | 46.0 | 33.7 | 23.7 | 15.2 | 9.5 | 3.6 | - |
| PGD-AT at $\epsilon = 0.034$ | 46.7 | 34.8 | 24.7 | 16.2 | 10.1 | 3.7 | 0.75 |
| PGD-AT at $\epsilon = 0.037$ | 44.9 | 33.3 | 23.7 | 15.6 | 10.0 | 3.8 | -0.07 |
| UDR-PGD at $\epsilon = 0.031$ | 48.9 | 36.4 | 26.3 | 18.5 | 13.0 | 7.1 | 3.08 |

robustness version (UDR-AWP-AT) also improves its counterpart by 0.5%. With the same setting (i.e., same architecture and without additional data), our UDR-TRADES and UDR-AWP-AT achieve better robustness than recently listed methods on RobustBench (Croce et al., 2020).[3] Remarkably, the additional experiment with C&W (L2) attack shows a significant improvement of our distributional methods over standard AT by around 5%. More discussion can be found in Appendix F.

## 5.2 ANALYTICAL RESULTS

**Benefit of the soft-ball projection.**   Here we would like to analytically study why our UDR methods are better than standard AT methods, by taking UDR-PGD and PGD-AT as examples. The visualization on the synthetic dataset can be found in Appendix E. Recall that one of the key differences between UDR-PGD and PGD-AT is that the former uses the soft-ball projection and the later use the hard-ball one, discussed in the second paragraph under Eq. (18). More specifically, Table 5 reports the average norm ($L_1$ and $L_\infty$) of the perturbation $\delta = |x^a - x|_p$ in PGD and our UDR-PGD. It can be seen that: (i) At the beginning of the training process, there is no difference between the norms of the perturbations generated by PGD and our UDR-PGD. More specifically, most of the pixels lie on the edge of the hard-ball projection (i.e., $p(0.9\epsilon \leq \delta \leq \epsilon) = p(\delta \leq \epsilon) - p(\delta \leq 0.9\epsilon) > 80\%$). (ii) When our model converges, there are 77.9% pixels lying slightly beyond the hard-ball projection (i.e., $p(\delta > \epsilon)$). It is because our soft-ball projection can be adaptive based on the value of . This flexibility helps the adversarial examples reach a better local optimum of the prediction loss, therefore, benefits the adversarial training.

Next, we show that doing PGD adversarial training with larger $\epsilon$ cannot achieve the same defence performance as our methods with the soft-ball projection. We conduct more experiments with PGD-AT with $\epsilon = 0.034$ (the final when our model converages) and $\epsilon = 0.037$ to show that simply extending the hard-ball projection doesn't benefit adversarial training. More specifically, the average robustness improvement with $\epsilon = 0.034$ is 0.75%, while there is no improvement with $\epsilon = 0.037$.

**Parameter sensitivity of $\tau$.** Figure 1a and 1b show the our framework's sensitivity to $\tau$ on CIFAR10 under the PGD attack. It can be observed that overly small values of $\tau$ can hardly improve adversarial robustness while overly big values of $\tau$ may hurt the natural performance ($acc_{nat} = 68.7\%$ with $\tau = 1.0$). Empirically, we find that $\tau = 2\eta$ performs well in our experiments.

## 6 CONCLUSIONS

In this paper, we have presented a new unified distributional robustness framework for adversarial training, which unifies and generalizes standard AT approaches with improved adversarial robustness. By defining a new family of risk functions, our framework facilitates the development of the distributional robustness counterparts of the SOTA AT methods including PGD-AT, TRADES, MART and AWP. Moreover, we introduce a new cost function, which enables us to bridge the connections between standard AT methods and their distributional robustness counterparts and to show that the former ones can be viewed as the special cases of the later ones. Extensive experiments on the benchmark datasets including MNIST, CIFAR10, CIFAR100 show that our proposed algorithms are able to boost the model robustness against strong attacks with better generalization capacity.

---

[3]RobustBench reported a robust accuracy of 56.17% for AWP-TRADES version from Wu et al. (2020) which is higher than ours but might not be used as a reference.

## ACKNOWLEDGEMENT

This work was partially supported by the Australian Defence Science and Technology (DST) Group under the Next Generation Technology Fund (NGTF) scheme. The authors are grateful to the anonymous (meta) reviewers for their helpful comments.

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

## A  THEORETICAL DEVELOPMENT

**Theorem 1.** *With the cost function $\tilde{c}_{\mathcal{X}}$ defined as above , the optimization problem:*

$$\inf_{\theta, \lambda \geq 0} \left\{ \lambda \epsilon + \mathbb{E}_{\mathbb{P}} \left[ \sup_{x'} \left\{ g_\theta \left( x', x, y \right) - \lambda \tilde{c}_{\mathcal{X}} \left( x', x \right) \right\} \right] \right\} \tag{21}$$

*is equivalent to the optimization problem:*

$$\inf_{\theta} \mathbb{E}_{\mathbb{P}} \left[ \sup_{x' \in B_\epsilon(x)} g_\theta \left( x', x, y \right) \right]. \tag{22}$$

*Proof.* We need to prove that

$$\inf_{\lambda \geq 0} \left\{ \lambda \epsilon + \mathbb{E}_{\mathbb{P}} \left[ \sup_{x'} \left\{ g_\theta \left( x', x, y \right) - \lambda \tilde{c}_{\mathcal{X}} \left( x', x \right) \right\} \right] \right\} = \mathbb{E}_{\mathbb{P}} \left[ \sup_{x' \in B_\epsilon(x)} g_\theta \left( x', x, y \right) \right]. \tag{23}$$

By the definition of the cost function $\tilde{c}_{\mathcal{X}}$, the LHS of (23) can be rewritten as:

$$\min \left\{ \inf_{\lambda > 0} \left\{ \lambda \epsilon + \mathbb{E}_{\mathbb{P}} \left[ \sup_{x' \in B_\epsilon(x)} \left\{ g_\theta \left( x', x, y \right) - \lambda c_{\mathcal{X}} \left( x', x \right) \right\} \right] \right\}, \mathbb{E}_{\mathbb{P}} \left[ \sup_{x'} g_\theta \left( x', x, y \right) \right] \right\}. \tag{24}$$

Given any $\lambda > 0$ and $x' \in B_\epsilon(x)$, we have

$$\lambda \epsilon + g_\theta \left( x', x, y \right) - \lambda c_{\mathcal{X}} \left( x', x \right) = g_\theta \left( x', x, y \right) + \lambda \left( \epsilon - c_{\mathcal{X}} \left( x', x \right) \right) \geq \mathbb{E}_{\mathbb{P}} \left[ g_\theta \left( x', x, y \right) \right].$$

Hence, we arrive at

$$\lambda \epsilon + \sup_{x' \in B_\epsilon(x)} \left\{ g_\theta \left( x', x, y \right) - \lambda c_{\mathcal{X}} \left( x', x \right) \right\} \geq \sup_{x' \in B_e(x)} g_\theta \left( x', x, y \right).$$

$$\lambda \epsilon + \mathbb{E}_{\mathbb{P}} \left[ \sup_{x' \in B_\epsilon(x)} \left\{ g_\theta \left( x', x, y \right) - \lambda c_{\mathcal{X}} \left( x', x \right) \right\} \right] \geq \mathbb{E}_{\mathbb{P}} \left[ \sup_{x' \in B_e(x)} g_\theta \left( x', x, y \right) \right].$$

which follows that

$$\inf_{\lambda > 0} \left\{ \lambda \epsilon + \mathbb{E}_{\mathbb{P}} \left[ \sup_{x' \in B_\epsilon(x)} \left\{ g_\theta \left( x', x, y \right) - \lambda c_{\mathcal{X}} \left( x', x \right) \right\} \right] \right\}$$

$$\geq \mathbb{E}_{\mathbb{P}} \left[ \sup_{x' \in B_e(x)} \mathbb{E}_{\mathbb{P}} \left[ g_\theta \left( x', x, y \right) \right] \right]. \tag{25}$$

We now prove the inequality

$$\lim_{\lambda \to 0^+} \left\{ \lambda \epsilon + \mathbb{E}_{\mathbb{P}} \left[ \sup_{x' \in B_\epsilon(x)} \{ g_\theta(x', x, y) - \lambda c_{\mathcal{X}}(x', x) \} \right] \right\}$$
$$= \mathbb{E}_{\mathbb{P}} \left[ \sup_{x' \in B_e(x)} \mathbb{E}_{\mathbb{P}} \left[ g_\theta(x', x, y) \right] \right].$$

Take a sequence $\{\lambda_n\}_{n \geq 1} \to 0^+$. Given a feasible pair $(x, y)$, we define

$$f_n(x'; x, y) := g_\theta(x', x, y) + \lambda_n [\epsilon - c_{\mathcal{X}}(x', x)], \forall x' \in B_\epsilon(x).$$

It is evident that $f_n(x'; x, y)$ converges pointwise to $g_\theta(x', x, y)$ over the compact set $B_\epsilon(x)$. Therefore, $f_n(x'; x, y)$ converges uniformly to $g_\theta(x', x, y)$ on this set. This follows that

$$\forall \alpha > 0, \exists n_0 = n(\alpha) : |f_n(x'; x, y) - g_\theta(x', x, y)| < \alpha, \forall x' \in B_\epsilon(x), n \geq n_0.$$

Hence, we obtain for all $x' \in B_\epsilon(x)$ and $n \geq n_0$:

$$g_\theta(x', x, y) - \alpha < f_n(x'; x, y) < g_\theta(x', x, y) + \alpha.$$

This leads to the following for all $n \geq n_0$:

$$\sup_{x' \in B_\epsilon(x)} g_\theta(x', x, y) - \alpha \leq \sup_{x' \in B_\epsilon(x)} f_n(x'; x, y) \leq \sup_{x' \in B_\epsilon(x)} g_\theta(x', x, y) + \alpha.$$

Therefore, we obtain:

$$\lim_{n \to \infty} \sup_{x' \in B_\epsilon(x)} f_n(x'; x, y) = \sup_{x' \in B_\epsilon(x)} g_\theta(x', x, y)$$

for all feasible pairs $(x, y)$, which further means that

$$\lim_{n \to \infty} \mathbb{E}_{\mathbb{P}} \left[ \sup_{x' \in B_\epsilon(x)} f_n(x'; x, y) \right] = \mathbb{E}_{\mathbb{P}} \left[ \sup_{x' \in B_\epsilon(x)} g_\theta(x', x, y) \right],$$

or equivalently

$$\lim_{n \to \infty} \mathbb{E}_{\mathbb{P}} \left[ \lambda_n \epsilon + \mathbb{E}_{\mathbb{P}} \left[ \sup_{x' \in B_\epsilon(x)} \{ g_\theta(x', x, y) - \lambda_n c_{\mathcal{X}}(x', x) \} \right] \right] = \mathbb{E}_{\mathbb{P}} \left[ \sup_{x' \in B_\epsilon(x)} g_\theta(x', x, y) \right]. \tag{26}$$

Because Eq. (26) holds for every sequence $\{\lambda_n\}_{n \geq 1} \to 0^+$, we reach

$$\lim_{\lambda \to 0^+} \left\{ \lambda \epsilon + \mathbb{E}_{\mathbb{P}} \left[ \sup_{x' \in B_\epsilon(x)} \{ g_\theta(x', x, y) - \lambda c_{\mathcal{X}}(x', x) \} \right] \right\}$$
$$= \mathbb{E}_{\mathbb{P}} \left[ \sup_{x' \in B_e(x)} \mathbb{E}_{\mathbb{P}} \left[ g_\theta(x', x, y) \right] \right]. \tag{27}$$

By combining (25) and (27), we reach

$$\inf_{\lambda > 0} \left\{ \lambda \epsilon + \mathbb{E}_{\mathbb{P}} \left[ \sup_{x' \in B_\epsilon(x)} \{ g_\theta(x', x, y) - \lambda c_{\mathcal{X}}(x', x) \} \right] \right\}$$
$$= \mathbb{E}_{\mathbb{P}} \left[ \sup_{x' \in B_e(x)} \mathbb{E}_{\mathbb{P}} \left[ g_\theta(x', x, y) \right] \right]. \tag{28}$$

Finally, we have

$$\inf_{\lambda \geq 0} \left\{ \lambda \epsilon + \mathbb{E}_{\mathbb{P}} \left[ \sup_{x'} \left\{ g_\theta \left( x', x, y \right) - \lambda \tilde{c}_{\mathcal{X}} \left( x', x \right) \right\} \right] \right\}$$

$$= \min \left\{ \inf_{\lambda > 0} \left\{ \lambda \epsilon + \mathbb{E}_{\mathbb{P}} \left[ \sup_{x' \in B_\epsilon(x)} \left\{ g_\theta \left( x', x, y \right) - \lambda c_{\mathcal{X}} \left( x', x \right) \right\} \right] \right\}, \mathbb{E}_{\mathbb{P}} \left[ \sup_{x'} g_\theta \left( x', x, y \right) \right] \right\}$$

$$= \min \left\{ \mathbb{E}_{\mathbb{P}} \left[ \sup_{x' \in B_e(x)} \mathbb{E}_{\mathbb{P}} \left[ g_\theta \left( x', x, y \right) \right] \right], \mathbb{E}_{\mathbb{P}} \left[ \sup_{x'} g_\theta \left( x', x, y \right) \right] \right\}$$

$$= \mathbb{E}_{\mathbb{P}} \left[ \sup_{x' \in B_e(x)} \mathbb{E}_{\mathbb{P}} \left[ g_\theta \left( x', x, y \right) \right] \right].$$

That concludes our proof. □

One of most technical challenge we need to bypass in our work is that in theory developed in Blanchet & Murthy (2019), to equivalently transform the primal form to the dual form, it requires the cost function to be finite. In the following theorem, we reprove the equivalence of the primal and dual forms in our context.

**Theorem 2.** *Assume that the function $g$ is upper-bounded by a number $L$. We have the following equality between the primal form and dual form*

$$\sup_{\mathbb{Q}: \mathcal{W}_c(\mathbb{Q}, \mathbb{P}_\triangle) < \epsilon} \mathbb{E}_{\mathbb{Q}} \left[ g \left( z' \right) \right] = \inf_{\lambda \geq 0} \left\{ \lambda \epsilon + \mathbb{E}_{\mathbb{P}_\triangle} \left[ \sup_{z'} \left\{ g \left( z' \right) - \lambda c \left( z', z \right) \right\} \right] \right\},$$

*where $z = (x, x, y)$, $z' = (x', x'', y')$, and we have defined*

$$c \left( z, z' \right) = \tilde{c}_{\mathcal{X}} \left( x, x' \right) + \infty \times \tilde{c}_{\mathcal{X}} \left( x, x'' \right) + \infty \times \mathbf{1} \left\{ y \neq y' \right\},$$

*for which we have defined*

$$\tilde{c}_{\mathcal{X}} \left( x, x' \right) = \begin{cases} c_{\mathcal{X}} \left( x, x' \right) & if\, c_{\mathcal{X}}(x, x') \leq \epsilon \\ \infty & otherwise. \end{cases}$$

*Proof.* Given a positive integer number $n > 0$, we define the following metrics:

$$c^n \left( z, z' \right) = \tilde{c}_{\mathcal{X}}^n \left( x, x' \right) + \infty \times \tilde{c}_{\mathcal{X}}^n \left( x, x'' \right) + \infty \times \mathbf{1} \left\{ y \neq y' \right\},$$

$$\tilde{c}_{\mathcal{X}}^n \left( x, x' \right) = \begin{cases} c_{\mathcal{X}} \left( x, x' \right) & if\, c_{\mathcal{X}} \left( x, x' \right) < \epsilon. \\ n & otherwise. \end{cases}$$

We have $\tilde{c}_{\mathcal{X}}^n \nearrow \tilde{c}_{\mathcal{X}}$ and $c^n \nearrow c$. We now prove that

$$\sup_{\mathbb{Q}: \mathcal{W}(\mathbb{Q}, \mathbb{P}_\triangle) < \epsilon} \mathbb{E}_{\mathbb{Q}} \left[ g \left( z' \right) \right] = \inf_n \sup_{\mathbb{Q}: \mathcal{W}_{c^n}(\mathbb{Q}, \mathbb{P}_\triangle) < \epsilon} \mathbb{E}_{\mathbb{Q}} \left[ g \left( z' \right) \right].$$

In fact, for each $n$, we have $c^n \leq c$. Therefore, $\mathcal{W}_{c^n} \left( \mathbb{Q}, \mathbb{P}_\triangle \right) \leq \mathcal{W}_c \left( \mathbb{Q}, \mathbb{P}_\triangle \right)$, hence $\{ \mathbb{Q} : \mathcal{W}_c \left( \mathbb{Q}, \mathbb{P}_\triangle \right) < \epsilon \} \subset \{ \mathbb{Q} : \mathcal{W}_{c^n} \left( \mathbb{Q}, \mathbb{P}_\triangle \right) < \epsilon \}$, implying that

$$\sup_{\mathbb{Q}: \mathcal{W}(\mathbb{Q}, \mathbb{P}_\triangle) < \epsilon} \mathbb{E}_{\mathbb{Q}} \left[ g \left( z' \right) \right] \leq \sup_{\mathbb{Q}: \mathcal{W}_{c^n}(\mathbb{Q}, \mathbb{P}_\triangle) < \epsilon} \mathbb{E}_{\mathbb{Q}} \left[ g \left( z' \right) \right].$$

$$\sup_{\mathbb{Q}: \mathcal{W}(\mathbb{Q}, \mathbb{P}_\triangle) < \epsilon} \mathbb{E}_{\mathbb{Q}} \left[ g \left( z' \right) \right] \leq \inf_n \sup_{\mathbb{Q}: \mathcal{W}_{c^n}(\mathbb{Q}, \mathbb{P}_\triangle) < \epsilon} \mathbb{E}_{\mathbb{Q}} \left[ g \left( z' \right) \right].$$

Let us define

$$A = \cup_{(x,y) \in \mathcal{D}} \left\{ (z, z') : z = (x, x, y), z' = (x', x'', y'), c_{\mathcal{X}} \left( x, x' \right) < \epsilon, x'' = x, y' = y \right\},$$

$$B = \cup_{(x,y) \in \mathcal{D}} \left\{ (z, z') : z = (x, x, y), z' = (x', x'', y'), c_{\mathcal{X}} \left( x, x' \right) \geq \epsilon, x'' = x, y' = y \right\}.$$

To simplify our proof, without generalization ability, for each $n$, we denote $\mathbb{Q}_n$ as the distribution in $\{\mathbb{Q} : \mathcal{W}_{c^n}(\mathbb{Q}, \mathbb{P}_\triangle) < \epsilon\}$ that peaks $\mathbb{E}_\mathbb{Q}[g_\theta(z')]$ and $\gamma_n$ as the optimal transport plan of $\mathcal{W}_{c^n}(\mathbb{Q}_n, \mathbb{P}_\triangle)$ which admits $\mathbb{P}_\triangle$ and $\mathbb{Q}_n$ as its marginals. Note that because $\mathcal{W}_{c^n}(\mathbb{Q}_n, \mathbb{P}_\triangle) < \epsilon$, the support of $\gamma_n$ almost surely determines on $A \cup B$. We then have

$$
\begin{aligned}
\mathcal{W}_{c^n}(\mathbb{Q}_n, \mathbb{P}_\triangle) &= \int c^n(z, z') \, d\gamma_n(z, z') \\
&= \int_A c^n(z, z') \, d\gamma_n(z, z') + \int_B c^n(z, z') \, d\gamma_n(z, z') \\
&= \int_A c_\mathcal{X}(x, x') \, d\gamma_n(z, z') + \int_B n \, d\gamma_n(z, z') \\
&= \int_A c_\mathcal{X}(x, x') \, d\gamma_n(z, z') + n\gamma_n(B) < \epsilon.
\end{aligned}
$$

Therefore, we obtain: $\gamma_n(B) < \frac{\epsilon}{n}$. We now define $\bar{\gamma}_n$ as a restricted measure of $\gamma_n$ on $A$, meaning that $\bar{\gamma}_n(C) = \frac{\gamma_n(A) + \gamma_n(B)}{\gamma_n(A)} \gamma_n(C) = (1 + o(n^{-1})) \gamma_n(C)$ for any measure set $C \subset A$, where $\lim_{n\to\infty} o(n^{-1}) = 0$. Let $\mathbb{P}_n$ as marginal distribution of $\mathbb{Q}_n$ corresponding to the dimensions of $z'$. It appears that

$$
\begin{aligned}
\mathcal{W}_c(\mathbb{P}_n, \mathbb{P}_\triangle) &\leq \int_A c(z, z') \, d\bar{\gamma}_n(z, z') + \int_B c(z, z') \, d\bar{\gamma}_n(z, z') \\
&\overset{(1)}{=} \int_A c_\mathcal{X}(x, x') \, d\bar{\gamma}_n(z, z') < \int_A \epsilon \, d\bar{\gamma}_n(z, z') = \epsilon.
\end{aligned}
$$

Note that we have $\overset{(1)}{=}$ because $\bar{\gamma}_n(B) = 0$.

This implies that $\mathbb{P}_n \in \{\mathbb{Q} : \mathcal{W}_c(\mathbb{Q}, \mathbb{P}_\triangle) < \epsilon\}$, which follows that

$$
\begin{aligned}
\sup_{\mathbb{Q}:\mathcal{W}(\mathbb{Q},\mathbb{P}_\triangle)<\epsilon} \mathbb{E}_\mathbb{Q}[g(z')] &\geq \mathbb{E}_{\mathbb{P}_n}[g_\theta(z')] = \mathbb{E}_{\bar{\gamma}_n}[g(z')] \\
&= \int_A g(z') \, d\bar{\gamma}_n(z, z') + \int_B g(z') \, d\bar{\gamma}_n(z, z') \\
&\overset{(1)}{=} \int_A g(z') \, d\bar{\gamma}_n(z, z') = \frac{\gamma_n(A) + \gamma_n(B)}{\gamma_n(A)} \int_A g(z') \, d\gamma_n(z, z') \\
&= (1 + o(n^{-1})) \left[ \int_{A \cup B} g(z') \, d\gamma_n(z, z') - \int_B g(z') \, d\gamma_n(z, z') \right] \\
&= (1 + o(n^{-1})) \left[ \int_{A \cup B} g(z') \, d\mathbb{Q}_n(z') - \int_B g(z') \, d\gamma_n(z, z') \right] \\
&\geq (1 + o(n^{-1})) \left[ \sup_{\mathbb{Q}:\mathcal{W}_{c^n}(\mathbb{Q},\mathbb{P}_\triangle)<\epsilon} \mathbb{E}_\mathbb{Q}[g_\theta(z')] - \int_B L \, d\gamma_n(z, z') \right]
\end{aligned}
$$

$$
\begin{aligned}
\sup_{\mathbb{Q}:\mathcal{W}(\mathbb{Q},\mathbb{P}_\triangle)<\epsilon} \mathbb{E}_\mathbb{Q}[g(z')] &\geq (1 + o(n^{-1})) \left[ \sup_{\mathbb{Q}:\mathcal{W}_{c^n}(\mathbb{Q},\mathbb{P}_\triangle)<\epsilon} \mathbb{E}_\mathbb{Q}[g_\theta(z')] - L\gamma_n(B) \right] \\
&\overset{(2)}{\geq} (1 + o(n^{-1})) \left[ \sup_{\mathbb{Q}:\mathcal{W}_{c^n}(\mathbb{Q},\mathbb{P}_\triangle)<\epsilon} \mathbb{E}_\mathbb{Q}[g_\theta(z')] - \frac{L\epsilon}{n} \right].
\end{aligned}
$$

Note that we have $\overset{(1)}{=}$ due to $\bar{\gamma}_n(B) = 0$ and $\overset{(2)}{\geq}$ due to $\gamma_n(B) < \frac{\epsilon}{n}$. Therefore, we reach the conclusion

$$
\sup_{\mathbb{Q}:\mathcal{W}(\mathbb{Q},\mathbb{P}_\triangle)<\epsilon} \mathbb{E}_\mathbb{Q}[g_\theta(z')] = \inf_n \sup_{\mathbb{Q}:\mathcal{W}_{c^n}(\mathbb{Q},\mathbb{P}_\triangle)<\epsilon} \mathbb{E}_\mathbb{Q}[g_\theta(z')].
$$

Next, we apply primal-dual form in Blanchet & Murthy (2019) for the finite metric $\tilde{c}^n_{\mathcal{X}}$ to reach

$$\sup_{\mathcal{W}_{c^n}(\mathbb{Q},\mathbb{P}_\triangle)<\epsilon} \mathbb{E}_\mathbb{Q}\left[g_\theta\left(z'\right)\right] = \inf_{\lambda\geq 0}\left\{\lambda\epsilon + \mathbb{E}_{\mathbb{P}_\triangle}\left[\sup_{z'}\left\{g_\theta\left(z'\right)-\lambda c^n\left(z',z\right)\right\}\right]\right\}.$$

Finally, taking $n \to \infty$ and noting that $c^n \nearrow c$, we reach the conclusion. $\square$

## B    FURTHER EXPLANATION WHY OUR UDR CAN UTILIZE GLOBAL INFORMATION AND THE ADVANTAGE OF SOFT-BALL

---

**Algorithm 1** The pseudocode of our proposed method.

---

**Input**: training set $\mathcal{D}$, number of iterations $T$, batch size $N$, adversary parameters $\{k,\epsilon,\eta\}$

**for** $t = 1$ to $T$ **do**

    1. **Sample** mini-batch $\{x_i,y_i\}_{i=1}^N \sim \mathcal{D}$

    2. **Find** adversarial examples $\{x_i^a\}_{i=1}^N$ using Eq. (18)

        (a) **Initialize** randomly: $x_i^0 = x_i + noise$ where $noise \sim \mathcal{U}(-\epsilon,\epsilon)$

        (b) **for** $n = 1$ to $k$ **do**

            i. $x_i^{inter} = x_i^n + \eta\mathrm{sign}\left(\nabla_x g_\theta(x_i^n,x_i,y_i)\right)$

            ii. $x_i^{n+1} = x_i^{inter} - \eta\lambda\nabla_x\hat{c}(x_i^{inter},x_i)$

        (c) **Clip** to valid range: $x_i^a = clip(x_i^k,0,1)$

    3. **Update** parameter $\lambda$ using Eq. (19)

    4. **Update** model parameter $\theta$ using Eq. (20)

**Output:** model parameter $\theta$

---

The advantage of our soft ball comes from the adaptive capability of $\lambda$, which is controlled by a global effect regarding how far adversarial examples $x_i^a$ from benign examples $x_i$. Let us revisit Algorithm 1. In the step 2.(b).i, we update

$$x_i^{inter} = x_i^n + \eta\mathrm{sign}\left(\nabla_x g_\theta(x_i^n,x_i,y_i)\right)$$

with the aim to find $x_i^{inter}$ that can maximize $g_\theta(\cdot,x_i,y_i)$ as in the standard versions.

Furthermore, in the step 2.(b).ii, we update

$$\begin{aligned}x_i^{n+1} &= x_i^{inter} - \eta\lambda\nabla_x\hat{c}(x_i^{inter},x_i) = x_i^{inter} - \eta\lambda\left(x_i^{inter}-x_i\right)\\&= (1-\eta\lambda)\,x_i^{inter} + \eta\lambda x_i,\end{aligned}\tag{29}$$

where we assume L2 cost $c(x,x') = \frac{1}{2}\|x-x'\|^2$ is used. It is evident that $x_i^{n+1}$ is an interpolation point of $x_i^{inter}$ and $x_i$, hence $x_i^{n+1}$ is drawn back to $x_i$ wherein the drawn-back amount is proportional to $\eta\lambda$.

We now revisit the formula to update $\lambda$ as Eq. (19)

$$\lambda_n = \lambda - \eta_\lambda\left(\epsilon - \frac{1}{N}\sum_{i=1}^N \hat{c}_{\mathcal{X}}(x_i^a,x_i)\right),$$

which indicates that $\lambda$ is globally controlled. More specifically, if average distance from $x_i^a$ to $x_i$ (i.e., $\frac{1}{N}\sum_{i=1}^N \hat{c}_{\mathcal{X}}(x_i^a,x_i)$) is less than $\epsilon$ (i.e., adversarial examples are globally close to benign examples), $\lambda$ is adapted decreasingly. Linking with the formula in Eq. (29), in this case, $x_i^{n+1}$ gets back to $x_i$ less aggressively to maintain the distance between $x_i^a$ and $x_i$. Otherwise, adversarial examples are globally far from benign examples, $\lambda$ is adapted increasingly. In this case, $x_i^{n+1}$ gets back to $x_i$ more aggressively to reduce more the distance between $x_i^a$ and $x_i$.

## C  Related Work

**Adversarial Attacks.** In this paper, we are interested in image classification tasks and focus on the adversaries that add small perturbations to the pixels of an image to generate attacks based on gradients, which are the most popular and effective. FGSM (Goodfellow et al., 2015) and PGD (Madry et al., 2018) are the most representative gradient-based attacks and PGD is the most widely-used one, due to its effectiveness and simplicity. Recently, there are several variants of PGD that achieve improved performance, for example, Auto-Attack by ensembling PGD with other attacks (Croce & Hein, 2020a) and the B&B method (Brendel et al., 2019) by attacking with decision-based boundary initialized with PGD. Along with PGD, these attacks have been considered as benchmark attacks for adversarial robustness.

**Adversarial defenses.** Among various kinds of defense approaches, Adversarial Training (AT), originating in Goodfellow et al. (2015), has drawn the most research attention. Given its effectiveness and efficiency, many variants of AT have been proposed with (1) different types of adversarial examples (e.g., the worst-case examples as in Goodfellow et al. (2015) or most divergent examples as in Zhang et al. (2019)), (2) different searching strategies (e.g., non-iterative FGSM and Rand FGSM (Madry et al., 2018)), (3) additional regularizations (e.g., adding constraints in the latent space (Zhang & Wang, 2019; Bui et al., 2020; 2021a; Hoang et al., 2020)), and (4) different model architectures (e.g., activation function (Xie et al., 2020) or ensemble models (Pang et al., 2019; Bui et al., 2021b)).

**Distributional robustness.** There have been a few works attempting to connect DR with adversarial machine learning or improve adversarial robustness based on the ideas of DR (Sinha et al., 2018; Staib & Jegelka, 2017; Miyato et al., 2018; Zhang & Wang, 2019; Najafi et al., 2019; Levine & Feizi, 2020; Le et al., 2022; Thanh et al., 2022). A recent work of Dong et al. (2020) proposes a new AT algorithm by constructing a distribution over each data sample to model the adversarial examples around it, which is still in the category of pointwise adversary (Sinha et al., 2018) and has no relations to DR. Although its aim of enhancing adversarial robustness is visually related ours, its mythology is different from ours. Therefore, we consider Sinha et al. (2018); Staib & Jegelka (2017) as the most relevant ones to ours. Specifically, both works leverage the dual form of Wasserstein DR (Blanchet & Murthy, 2019) for searching worst-case perturbations for AT, where Sinha et al. (2018) (WRM) focuses on certified robustness with comprehensive study on the tradeoffs between complexity, generality, guarantees, and speed, while Staib & Jegelka (2017) (FDRO) points out that Wasserstein robust optimization can be viewed as the generalization to standard AT.

Although our study is inspired by the two works, there are significant differences and new results of ours: **1)** We introduce a new Wasserstein cost function and a new series of risk functions in WDR, which facilitate our framework to generalize and encompass many SOTA AT methods. While WRM can be viewed as the generalization to PGD-AT only. **2)** Most importantly, although WDR has been demonstrated to have superior properties over standard AT in the two papers, unfortunately, WRM and FDRO have not been observed to outperform standard AT methods. For example, the experiments of FDRO show that adversarial robustness on MNIST of WRM and FDRO is worse than that of AT with PGD and iterative-FGSM (Staib & Jegelka, 2017). Moreover, WRM and FDRO's effectiveness either on more complex colored images (e.g., CIFAR10) or against more advanced attacks (e.g., Auto-Attack) has not been carefully studied yet. On the contrary, we conduct extensive experiments to show the SOTA performance of our proposed algorithms.

## D  Experimental Settings

**For MNIST dataset.** We use a standard CNN architecture for the MNIST dataset which is identical with that in Carlini & Wagner (2017). We use the SGD optimizer with momentum 0.9, starting learning rate 1e-2 and reduce the learning rate ($\times 0.1$) at epoch {55, 75, 90}. We train with 100 epochs.

**For CIFAR10 and CIFAR100 dataset with ResNet18 architecture.** We use the ResNet18 for the CIFAR10 and CIFAR100 dataset. We use the SGD optimizer with momentum 0.9, weight decay 3.5e-3 as in the official implementation from Wang et al. (2019).[4] The starting learning rate 1e-2 and reduce the learning rate ($\times 0.1$) at epoch {75, 90, 100}. We train with 200 epochs.

---

[4]https://github.com/YisenWang/MART

**For hard/soft-ball projection experiments.** For PGD-AT, we use the following three ad-hoc strategies for $\epsilon$: 1) Fixing $\epsilon = 8/255$; 2) Fixing $\epsilon = 16/255$; 3) Gradually increasing/decreasing $\epsilon$ from $8/255$ to $16/255$, from epoch 20 to epoch 70, with the changing rate $\delta = 8/255/50$ per epoch. For example, the perturbation bound of the increasing strategy at epoch $i$ is: $\epsilon_i = \min(\frac{16}{255}, \max(\frac{8}{255}, \frac{8}{255} + (i - 20)\delta))$; the perturbation bound for decreasing strategy is: $\epsilon_i = \max(\frac{8}{255}, \min(\frac{16}{255}, \frac{16}{255} - (i - 20)\delta))$.

**For CIFAR10 with WideResNet architecture.** We follow the setting in Pang et al. (2020) for the additional experiments on CIFAR10 with WideResNet-34-10 architecture. More specifically, we train with 200 epochs with SGD optimizer with momentum 0.9, weight decay 5e-4. The learning rate is 0.1 and reduce at epoch 100th and 150th with rate 0.1 (Rice et al., 2020; Wu et al., 2020). More importantly, to match the performance as reported in Croce et al. (2020), we omit the cross-entropy loss of the natural images in PGD-AT and UDR-PGD. More specifically, the objective function of PGD-AT in Eq. (5) has been replaced by: $\inf_\theta \mathbb{E}_\mathbb{P} \left[ \beta \sup_{x' \in B_\epsilon(x)} CE\left(h_\theta\left(x'\right), y\right) \right]$ while the unified risk function for UDR-PGD to be: $g_\theta\left(z'\right) := \beta CE\left(h_\theta\left(x'\right), y'\right)$. We also switch Batch Normalization layer to evaluation stage when crafting adversarial examples as adviced in Pang et al. (2020).

# E VISUALIZING THE BENEFIT OF DISTRIBUTIONAL ROBUSTNESS

**Synthetic dataset setting.** We conduct an experiment on a synthetic dataset with a simple MLP model to visualize the benefit of our UDR framework over the standard AT methods, by taking UDR-PGD and PGD-AT as examples. The synthetic dataset consists of three clusters A, B1, B2 where A, B are two classes as shown in Figure 2c. The data points are sampled from normal distributions, i.e., $A \sim \mathcal{N}\left((-2, 0), \Sigma\right), B1 \sim \mathcal{N}\left((2, 0), \Sigma\right)$ and $B2 \sim \mathcal{N}\left((6, 0), \Sigma\right)$ where $\Sigma = 0.5 * I$ with $I$ is the identity matrix. There are total 10k training samples and 2k testing samples with densities of three clusters are 10%, 50% and 40%, respectively. We use a simple model of 4 Fully-Connected (FC) layers as follows: Input –> ReLU(FC(10)) –> ReLU(FC(10)) –> ReLU(FC(10)) –> Softmax(FC(2)), where FC(k) represents for FC with k hidden units. We use Adam optimizer with learning rate 1e-3 and train with 30 epochs. We use $\{k = 20, \epsilon = 1.0, \eta = 0.1\}$ for adversarial training (either PGD-AT or UDR-PGD) and PGD attack with $\{k = 200, \epsilon = 2.0, \eta = 0.1\}$ for evaluation.

It is a worth noting that while the distance between clusters is 2, we limit the perturbation $\epsilon = 1$ for the adversarial training to show the advantage on the flexibility of the soft-ball projection on the same/limited perturbation budget. Intuitively, cluster A has the lowest density (10%), therefore, the ideal decision boundary should be surrounded cluster A which sacrifices the robustness of the cluster A but increases the overall robustness eventually.

**Comparison between UDR-PGD and PGD-AT.** First, we visualize the trajectory of adversarial example from PGD and our UDR-PGD as in Figures 2b,2a to compare behaviors of two adversaries on the same pre-trained model. It can be seen that: (i) the PGD's adversarial examples and ours are pushed toward the lower confident region to maximize the prediction loss $g_\theta(x', x, y)$; (ii) however, while the adversarial examples of PGD are limited on the hard-projection ball, our adversarial examples have more flexibility. Specifically, those are close to the decision boundary (cluster A, B1) can go further, while those are distant to the decision boundary (cluster B2) stay close to the original input. This flexibility helps the adversarial examples reach better local optimum of the prediction loss, hence, benefits the adversarial training. Consequently, as shown in Figure 2c the final decision boundary of our UDR-PGD is closer to the ideal decision boundary than that of PGD-AT, hence, achieving a better robustness. Quantitative result shows that the robust accuracy of our UDR-PGD is 82.6%, while that of PGD-AT is 74.5% with the same PGD attack $\{k = 200, \epsilon = 2.0, \eta = 0.1\}$.

**Comparison among UDR-PGD settings.** Here we would like to provide more understanding about our framework through the experiment with PGD-AT as shown in Figure 3. First, we compare the trajectories of the adversarial examples of UDR-PGD with different $\lambda$ as shown in Figures 3a,3b. It can be seen that the crafted adversarial examples stay closer to their benign counterparts when $\lambda$ becomes higher (i.e., $\lambda = 0.1$ in Figure 3a). In contrast, the soft-projection ball is extended when $\lambda$ becomes smaller (i.e., $\lambda = 0.01$ in Figure 3b). On the other hand, with the same $\lambda$ but smaller $\tau$ as

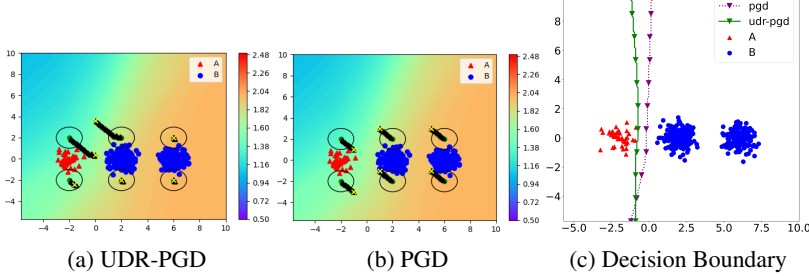

|                |              |                      |
| :------------: | :----------: | :------------------: |
| (a) UDR-PGD    | (b) PGD      | (c) Decision Boundary |

Figure 2: (a)/(b): Trajectory of PGD and UDR-PGD adversarial examples. Each trajectory includes 20 intermediate steps. For better visualization, we do not use random initialization. The model is the natural training at epoch 1. (c) The final decision boundary comparison.

shown in Figure 3c, the soft-ball projection is more identical to the hard ball projection as shown in Figure 2b. These behaviors concur with the theoretical expectation as discussed in Section 4.1 in the main paper.

Figure 3d shows the learning progress of parameter $\lambda$. It can be observed that (i) the $\lambda$ converges to 0 regardless of its initialization value and (ii) the convergence rate of $\lambda$ depends on the parameter $\tau$ (i.e., smaller $\tau$ slower convergence). We choose $\tau = 2\eta$ for the experiments on real-world image datasets.

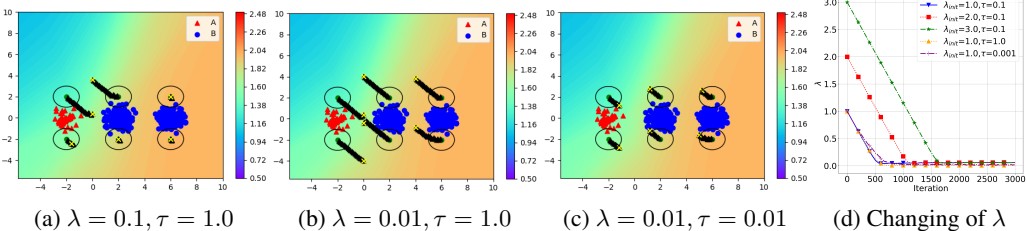

|                              |                                 |                                   |                       |
| :--------------------------: | :-----------------------------: | :-------------------------------: | :-------------------: |
| (a) $\lambda = 0.1, \tau = 1.0$ | (b) $\lambda = 0.01, \tau = 1.0$ | (c) $\lambda = 0.01, \tau = 0.01$ | (d) Changing of $\lambda$ |

Figure 3: (a)/(b)/(c): Trajectory of UDR-PGD adversarial examples with different settings. Each trajectory includes 20 intermediate steps. For better visualization, we do not use random initialization. The model is the natural training at epoch 1. (d) The changing of parameter $\lambda$.

**Further results of soft-ball projection.** In Figure 4, we compare our UDR-PGD with the soft-ball projection to PGD-AT with the hard-ball projection with different settings against the PGD attack on CIFAR10. For PGD-AT, we use the following three ad-hoc strategies for $\epsilon$: 1) Fixing $\epsilon = 8/255$; 2) Fixing $\epsilon = 16/255$; 3) Gradually increasing/decreasing $\epsilon$ from 8/255 to 16/255 (Refer to Appendix D for details). It can be seen that it is hard to find an effective strategy of the perturbation boundary of the hard-ball projection for PGD-AT, which can outperform ours. This demonstrates the benefit of our soft-project operation.

## F  MORE RESULTS AND ANALYSIS

**Further results with C&W (L2) attack.** We enrich the comprehensiveness of the experiments by further evaluating the defense methods with C&W (L2) attack (Carlini & Wagner, 2017) which is a very strong optimization based attack. The experiment has been conducted on the CIFAR10 dataset with WideResNet architecture. The hyper-parameters are $c \in \{0.5, 0.7, 1.0\}, kappa = 0, steps = 1000, lr = 0.01$ where $kappa$ is the confidence coefficient and $c$ is box-constraint coefficient.[5] As shown in Table 7, our distributional robustness version significantly outperform the standard ones in term of robust accuracy. For example, against C&W (c=0.5) attack, the robust accuracy gap between UDR-PGD and PGD-AT is $6\%$ while that for UDR-AWP-AT and AWP-AT is around $5\%$. The average improvement of robust accuracies against different levels of attack strengths is around

---

[5]We use the implementation from https://github.com/Harry24k/adversarial-attacks-pytorch

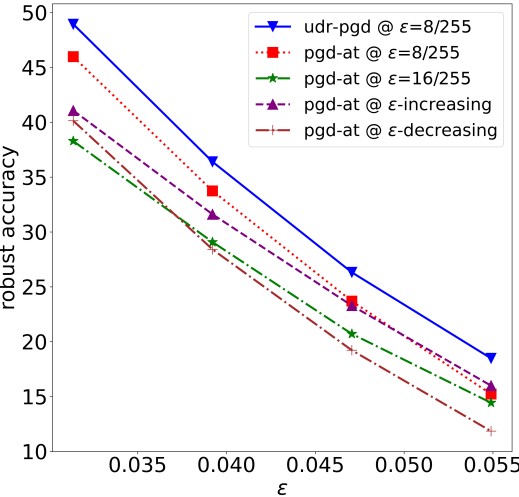

Figure 4: Hard/soft-ball projections

Table 7: Robustness evaluation against C&W attack with WRN-34-10 on the full test set of the CIFAR10 dataset (10K test images). $c$ is box-constraint coefficient. (*) Omit the cross-entropy loss of natural images.

|  | Nat | $c = 0.5$ | $c = 0.7$ | $c = 1.0$ | Avg-Gap |
|---|---|---|---|---|---|
| PGD-AT* | 84.93 | 40.85 | 25.90 | 12.95 | - |
| UDR-PGD* | 84.60 | **47.31** | **31.58** | **16.57** | 5.25 |
| TRADES | 85.70 | 47.65 | 34.30 | 21.03 | - |
| UDR-TRADES | 84.93 | **49.14** | **36.33** | **23.28** | 1.92 |
| AWP-AT | 85.57 | 49.91 | 34.31 | 18.97 | - |
| UDR-AWP-AT | 85.51 | **54.44** | **39.86** | **23.61** | 4.91 |

5%. This result strongly emphasizes the contribution of our distributional robustness and the soft-ball projection over the standard adversarial training.

**Experimental results of WRM (Sinha et al., 2018).** The performance of WRM highly depends on the Lagrange dual parameter $\gamma$ (or $\epsilon = 0.5/\gamma$ in their implementation[6]), which controls the robustness level. As mentioned in their paper, with large $\gamma$, the method is less robust but more tractable. Generally, decreasing $\gamma$ will reduce the natural accuracy but increase the robustness of the model as shown in Table 8. We obtained the best performance on MNIST with $\gamma = 0.05$ (CNN), while on CIFAR10 and CIFAR100 with $\gamma = 0.5$ (ResNet18). The best results with three benchmark datasets have been reported as in Table 9 (recall results from Table 1). It is a worth mentioning that while we could obtain a similar performance as reported Sinha et al. (2017) on the MNIST dataset with their architecture (3 Convolution layers + 1 FC layer), however, WRM seems much less effective with larger architectures.

Table 8: Result of WRM with different $\epsilon = 0.5/\gamma$ on the CIFAR10 dataset.

|  | Nat | PGD | AA | B&B |
|---|---|---|---|---|
| $\epsilon = 0.1$ | 90.9 | 15.3 | 13.7 | 15.8 |
| $\epsilon = 0.5$ | 86.7 | 33.9 | 32.6 | 35.4 |
| $\epsilon = 1.0$ | 83.7 | 40.9 | 39.8 | 41.4 |
| $\epsilon = 2.0$ | 79.4 | 45.4 | 43.6 | 45.5 |
| $\epsilon = 5.0$ | 71.6 | 47.5 | 45.2 | 46.2 |
| $\epsilon = 10.0$ | 65.0 | 46.6 | 43.4 | 44.4 |

---

[6]https://github.com/duchi-lab/certifiable-distributional-robustness/blob/master/attacks_tf.py

Table 9: Comparisons of natural classification accuracy (Nat) and adversarial accuracies against different attacks. Recall results from Table 1 with additional results of WRM. Best scores are highlighted in boldface.

| | MNIST | | | | CIFAR10 | | | | CIFAR100 | | | |
|---|---|---|---|---|---|---|---|---|---|---|---|---|
| | Nat | PGD | AA | B&B | Nat | PGD | AA | B&B | Nat | PGD | AA | B&B |
| WRM | 91.8 | 27.1 | 4.5 | 8.2 | 83.7 | 40.9 | 39.8 | 41.4 | 56.6 | 24.7 | 21.3 | 22.9 |
| PGD-AT | 99.4 | 94.0 | 88.9 | 91.3 | **86.4** | 46.0 | 42.5 | 44.2 | 72.4 | 41.7 | 39.3 | 39.6 |
| UDR-PGD | **99.5** | **94.3** | **90.0** | **91.4** | **86.4** | **48.9** | **44.8** | **46.0** | **73.5** | **45.1** | **41.9** | **42.3** |

**Further results of whitebox attacks with varied $\epsilon$.** Here we would like to provide more results on defending against whitebox attacks with a bigger range of $\epsilon$ as shown in Figure 5. It can be seen that in a wide range of attack strengths our DR methods consistently outperform their AT counterparts.

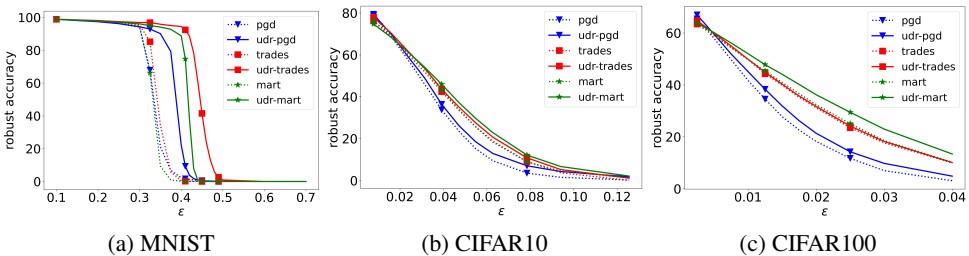

(a) MNIST  (b) CIFAR10  (c) CIFAR100

Figure 5: Robustness evaluation against multiple attack strengths.

**The convergence of the algorithm.** During the training, we observed that while adversarial examples distribute inside/outside the hard ball $\epsilon$ differently (i.e., as shown in Figure 2a ), but generally the average distance to original input is less than $\epsilon$. Therefore, according to the update formulation in Eq. (19), $\lambda$ tends to decrease to 0 and eventually is stable at 0 because of very small learning rate as shown in Figure 3d. In addition, we visualize the training progress as shown in Figure 6 to show the convergence of our method. It can be seen that, the error-rate reduces over training progress and converges at the end of the training progress.

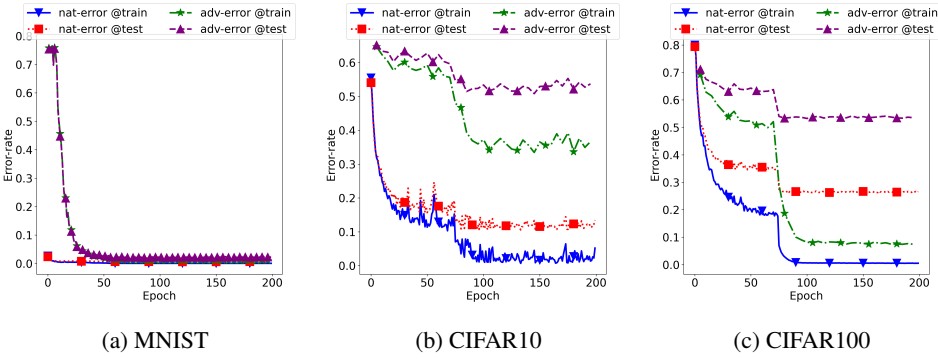

(a) MNIST  (b) CIFAR10  (c) CIFAR100

Figure 6: Training progress of our UDR-PGD on different datasets, evaluating on the full training set (e.g., 50k images) and the full testing set (e.g., 10k images). Robust accuracy is against PGD attack with $k = 20$.

**Further experiment result on CIFAR100.** We would like to provide additional experiment result on CIFAR100 dataset such that all defenses are adversarially trained with $\epsilon = \frac{8}{255}$. Our UDR-PGD outperforms PGD 3.7% at $\epsilon = \frac{8}{255}$ and 2.3% on average, while our UDR-TRADES and UDR-MART outperform their counterparts by around 0.5% and 0.7%, respectively. It is worth noting that, in our experiment, MART is quite sensitive with changes of (MART's natural accuracy drops to a lower

Table 10: Robustness evaluation on CIFAR100 dataset. The last column "Avg" represents the average gap of robust accuracy between our methods and their standard AT counterparts.

| | Nat | $\frac{8}{255}$ | $\frac{10}{255}$ | $\frac{12}{255}$ | $\frac{14}{255}$ | $\frac{16}{255}$ | $\frac{20}{255}$ | Avg |
|---|---|---|---|---|---|---|---|---|
| PGD-AT | 63.7 | 22.8 | 16.1 | 11.4 | 7.8 | 5.1 | 2.4 | - |
| UDR-PGD | 64.5 | 26.5 | 18.9 | 13.7 | 9.8 | 7.0 | 3.5 | 2.30 |
| TRADES | 60.2 | 30.3 | 24.5 | 18.8 | 14.8 | 11.5 | 6.7 | - |
| UDR-TRADES | 60.1 | 30.8 | 25.1 | 19.3 | 15.5 | 12.2 | 7.5 | 0.52 |
| MART | 54.1 | 32.0 | 26.8 | 21.9 | 17.4 | 13.8 | 7.6 | - |
| UDR-MART | 54.4 | 32.3 | 27.4 | 22.5 | 18.4 | 14.4 | 8.5 | 0.67 |

Table 11: Distance function and its gradient

| | $c_{\mathcal{X}}(x, x')$ | $\nabla_{x'} c(x, x')$ |
|---|---|---|
| $L_1$ | $\sum_{i=1}^{d} \left\| x_i - x_i' \right\|$ | $1, \forall i \in [1, d]$ |
| $L_2$ | $\frac{1}{2} \sum_{i=1}^{d} (x_i - x_i')^2$ | $\sum_{i=1}^{d} (x_i' - x_i)$ |
| $L_\infty$ | $\max_i \left\| x_i - x_i' \right\|$ | $\begin{cases} 1, i = \text{argmax}_i \left\| x_i - x_i' \right\| \\ 0, \text{otherwise} \end{cases}$ |

performance than that of TRADES); that might explain the lower gap between UDR-MART and MART with the new $\epsilon$.

## G  CHOOSING THE COST FUNCTION

In this section, we provide the technical details of our learning algorithm in Section 4 in the main paper, especially, the important of choosing cost function $\hat{c}(x, x')$. Given the current model $\theta$ and the parameter $\lambda$, we find the adversarial examples by solving:

$$x^a = \text{argmax}_{x'} \left\{ g_\theta(x', x, y) - \lambda \hat{c}_{\mathcal{X}}(x', x) \right\}$$

We employ multiple gradient ascent update steps without projecting onto the hard ball $B_\epsilon$. Specifically, the updated adversarial at step $t + 1$ as follows:

$$x^{t+1} = x^t + \eta \left( \nabla_{x'} g_\theta(x', x, y) - \lambda \nabla_{x'} \hat{c}_{\mathcal{X}}(x', x) \right)$$

Given the smoothed cost function as in Equation (19), the updating step is as follows:

$$x^{t+1} = \begin{cases} x^t + \eta \left( \nabla_{x'} g_\theta(x', x, y) - \lambda \nabla_{x'} c_{\mathcal{X}}(x', x) \right), & \text{if } c_{\mathcal{X}}(x', x) < \epsilon \\ x^t + \eta \left( \nabla_{x'} g_\theta(x', x, y) - \frac{\lambda}{\tau} \nabla_{x'} c_{\mathcal{X}}(x', x) \right), & \text{otherwise.} \end{cases}$$

It shows that, the pixels that are out-of-perturbation ball $B_\epsilon$ will be traced back with a longer step, depending on the parameter $\tau$. We consider three popular distance functions of $c_{\mathcal{X}}(x', x)$ with their gradient as Table 11. It is worth noting that, while the norm $L_1, L_2$ have gradient in all pixels, the $L_\infty$ has gradient in only one pixel per image. It means that, when using $L_\infty$ norm as the cost function $c_{\mathcal{X}}(x, x')$, only single pixel has been traced back at each iteration. In contrast, using $L_2$ will project all pixels toward the original input $x$ with the step size of each. As in the discussion in Section F, only small part of an MNIST image contributes to the prediction, while in contrast, most of pixels of a CIFAR10 image affect to the prediction. Based on this observation, we use the $L_\infty$ for the MNIST dataset and $L_2$ for the CIFAR10 dataset in the updating step. However, the perturbation strength $\epsilon$ has been measured in $L_\infty$, therefore, we still use $L_\infty$ in the Equation (22) to update $\lambda$.

We also visualize the histogram of gradient $\nabla_{x'} g_\theta(x', x, y)$ and $\nabla_{x'} \hat{c}_{\mathcal{X}}(x', x)$ as shown in Figure 7. It can be seen that the strength of gradient $grad1 = \nabla_{x'} g_\theta(x', x, y)$ is much smaller than $grad2 = \nabla_{x'} \hat{c}_{\mathcal{X}}(x', x)$, for example, on the MNIST dataset, $grad1 \in [-5 \times 10^{-4}, 5 \times 10^{-4}]$ while $grad2 \in [-0.3, 0.3]$ which is 600 times larger. Therefore, if using single update step, the gradient $\nabla_{x'} \hat{c}_{\mathcal{X}}(x', x)$ dominates the other and pulls the adversarial examples close to the natural input. These adversarial examples are weaker and do not helps to improve the robustness. Alternatively,

we break single update step for solving Equation (21) to two sub-steps as shown in Algorithm 1 to balance between push/pull steps. It also can be seen that the $grad2$ corresponds with the perturbation boundary $\epsilon$ and the step size $\eta$. For example, on the MNIST dataset, $grad2$ has the range from $[-0.3, 0.3]$ and has the highest density around $[-0.01, 0.01]$ where $\{0.3, 0.01\}$ are the perturbation boundary and step size in the experiment.

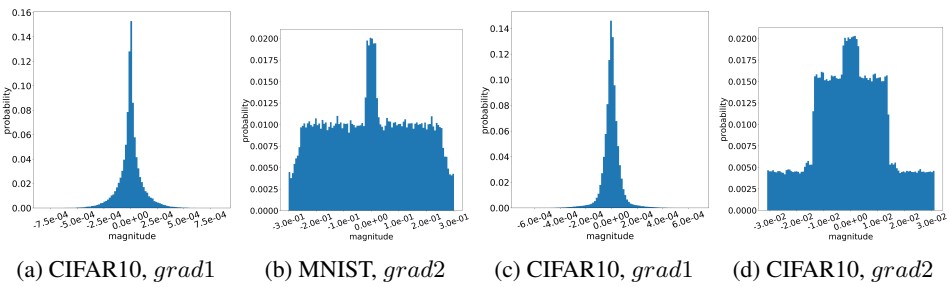

(a) CIFAR10, $grad1$   (b) MNIST, $grad2$   (c) CIFAR10, $grad1$   (d) CIFAR10, $grad2$

Figure 7: Histogram of gradient strength of $grad1 = \nabla_{x'} g_\theta(x', x, y)$ and $grad2 = \nabla_{x'} \hat{c}_\mathcal{X}(x', x)$ on MNIST and CIFAR10 dataset. We use $L_2$ norm for the cost function $c_\mathcal{X}(x', x)$, $\tau = \eta$ and $\lambda = 1$

