# OpenReview forum: "A Unified Wasserstein Distributional Robustness Framework for Adversarial Training"
_ICLR.cc/2022/Conference — ICLR 2022 Poster_

### Official Review · Reviewer_Rag4 · 2021-10-31

**Correctness:** 3
**Technical Novelty And Significance:** 3
**Empirical Novelty And Significance:** 3
**Recommendation:** 6
**Confidence:** 3

**Main Review:**

Strengths:

The unified distributional robustness framework presents a different perspective to adversarial robustness of deep learning through new Wasserstein cost functions. This work improves WRM (Sinha et al., 2017) to cover more adversarial training methods, e.g., TRADES and MART. This paper also develops a new family of algorithms that generalize the AT methods in the standard robustness setting, and experimental results show the proposed algorithms achieve better performance than standard AT methods.

Weakness:
1.	The theoretical contribution (Theorem 1) needs to be better justified. It is not very clear how this work differs from WRM (Sinha et al., 2017) with respect to the main conclusions of equivalence and proof techniques. From Theorem 1, it does not seem obvious that “the standard PGD-AT, TRADES, and MART are special cases of their UDR counterparts.” It is also unclear how significant the boundness of cost function is, although the extension of results from bounded to unbounded cost function may be not trivial.
2.	The literature review does not seem comprehensive – some recent distributional robustness methods are not mentioned, e.g., feature scattering (Zhang & Wang, 2019), Adversarial Distributional Training (Dong et al., 2020)
3.	When using different SOTA attacks to evaluate the defense methods, why the C&W attack (Carlini & Wagner, 2017) is excluded from the comparison?
4.	Some typos here and there. For example, “The see why the later is the case, …”; “… and a new serious of risk functions in WDR, …”; “While when τ is set closer to 0, …”; “The results of this experiment is shown …”; “… are different from those the model is trained with”; “…, and the later use the hard-ball one, …”; “… therefore, benefits the adversarial training.”; “doing PGD adversarial training with larger cannot achieve …”


**Summary Of The Paper:**

This work aims to improve WRM (Sinha et al., 2017) with respect to Wasserstein distributional robustness to cover more state-of-the-art adversarial training methods in addition to the most commonly used PGD-AT method. To this end, this paper presents a unified framework to connect Wasserstein distributional robustness with PGD-AT, TRADES, and MART methods. The introduced Wasserstein cost function can cover those of commonly used AT methods as special cases. A family of algorithms are developed to generalize AT methods with better generalization. Extensive experiments show the improvement of the proposed robustness AT algorithms over those commonly used ones.

**Summary Of The Review:**

In general, this paper presents some solid contributions to distributional robustness including both theoretical and experimental results. This paper is acceptable to the conference, but some weaknesses should be fixed, including the theoretical justifications and more comprehensive comparisons.

---

> ### Author Response · Authors · 2021-11-19
> **Response to Reviewer Rag4**
>
> We thank the Reviewer for your positive comments. Please find our response as below:
>
> **1. The theoretical contribution (Theorem 1) needs to be better justified.
> It is not very clear how this work differs from WRM (Sinha et al., 2017) with respect
> to the main conclusions of equivalence and proof techniques. From Theorem 1, it does
> not seem obvious that “the standard PGD-AT, TRADES, and MART are special cases of
> their UDR counterparts.” It is also unclear how significant the boundness of cost
> function is, although the extension of results from bounded to unbounded cost
> function may be not trivial.**
>
> Our work is developed based on Blanchet \&Murthy (2019) which is different from Sinha et al.  (2017). Please refer to Section 2.1 in the main paper for the discussion of the difference of Blanchet \&Murthy (2019) and Sinha et al.  (2017). The primal form of Sinha et al.  (2017) is the left side of Eq. (4), while its dual form is the right side of Eq. (4). Meanwhile, the primal form of Blanchet \&Murthy (2019) is Eq. (1), while its dual form is Eq. (3). The key difference of two works is that in Sinha et al.  (2017), $\lambda$ is a predefined hyper-parameter, while in Blanchet \&Murthy (2019), $\lambda$ is learnable.
>
> Regarding our contributions, we first propose a unified and novel viewpoint to enrich expressiveness ability of Blanchet \&Murthy (2019) allowing us to present most of popular adversarial training approaches. We note that in Sinha et al.  (2017), the function of interest is only the loss function as shown in Eq. (9). By devising specific extended space and distribution $\mathbb{P}_\Delta$, we demonstrate that we can really enrich the framework in Blanchet \&Murthy (2019) for capturing all of popular standard adversarial training approaches.
>
> Therefore, each standard adversarial training approach has its own counterpart in our framework. To further make connection between standard adversarial training approaches and their counterparts in our framework, we prove that for the specific cost function defined in Eq. (13), the counterparts in our framework reduce exactly to their corresponding standard adversarial training approaches. For this purpose, there are two theoretical obstacles we need to bypass. First,  the transformation from primal to dual forms in Eq. (11) requires the cost function $c$ to be bounded. In Theorem 2 in Appendix A, we prove this
> primal-dual form transformation for the unbounded cost function $\tilde{c}_X$, which is certainly not trivial. Second, after gaining the dual form, in Theorem 1, we need to prove the equivalence of the standard adversarial approaches and their counterparts for the specific cost function $\tilde{c}_X$.
>
> **2. The literature review does not seem comprehensive – some recent
> distributional robustness methods are not mentioned, e.g., feature scattering
> (Zhang \& Wang, 2019), Adversarial Distributional Training (Dong et al., 2020)**
>
> We have discussed these related works in our revision.
>
> **3. When using different SOTA attacks to evaluate the defense methods,
> why the C\&W attack (Carlini \& Wagner, 2017) is excluded from the comparison?**
>
> We have evaluated the defense methods with C\&W (L2) attack as your suggestion and would like to
> report the new result as Table below. We use the implementation from [1] with hyper-parameter
> setting as $c\in \{0.5,0.7,1.0\}, kappa=0, steps=1000, lr=0.01$ where $kappa$ is the confidence coefficient
> and $c$ is box-constraint coefficient. Surprisingly, our distributional robustness version significantly
> outperform the standard ones in term of robust accuracy. For example, against C\&W (c=0.5) attack,
> the robust accuracy gap between UDR-PGD and PGD-AT is $6\%$  while that for UDR-AWP-AT and AWP-AT
> is around $5\%$. The average improvement of robust accuracies against different levels of attack strengths is around $5\%$. This result strongly emphasizes the contribution of our distributional robustness and the soft-ball projection over the standard adversarial training.
>
> |            |  Nat  | c=0.5 | c=0.7 | c=1.0 | Avg-Gap |
> |------------|:-----:|:-----:|:-----:|:-----:|:-------:|
> | PGD-AT*    | 84.93 | 40.85 | 25.90 | 12.95 |    -    |
> | UDR-PGD*   | 84.60 | 47.31 | 31.58 | 16.57 |   5.25  |
> | TRADES     | 85.70 | 47.65 | 34.30 | 21.03 |    -    |
> | UDR-TRADES | 84.93 | 49.14 | 36.33 | 23.28 |   1.92  |
> | AWP-AT     | 85.57 | 49.91 | 34.31 | 18.97 |    -    |
> | UDR-AWP-AT | 85.51 | 54.44 | 39.86 | 23.61 |   4.91  |
>
> [1] https://github.com/Harry24k/adversarial-attacks-pytorch
>
> **4. Comments on the writing**
>
> Thanks, we have updated the paper based on your suggestions.

---

### Official Review · Reviewer_1ivF · 2021-11-02

**Correctness:** 3
**Technical Novelty And Significance:** 3
**Empirical Novelty And Significance:** 3
**Recommendation:** 6
**Confidence:** 3

**Main Review:**

### Theoretical contribution
The paper extends the previously established connection between AT, distributional robustness and optimal transport (Blanchet & Murthy [2016 on arxiv, 2019 published]; Sinha et al. 2017). However, my impression is that the main theoretical contribution for employing the work by Blanchet et al. for connecting distributional robustness with AT was made by Sinha et al. 2017: They proved a relaxed version of the dual form of the risk function using a Lagrangian penalty formulation and show the way to place PGD-AT in this WDR framework.
The paper proposed here extends this result to incorporate other AT approaches in the relaxed dual form established by Sinha et al. 2017. To achieve this, the authors reformulate the cost function proposed by Sinha et al. 2017 such that it incorporates TRABES and MART as well. As a consequence their cost function becomes unbounded (eq 13), which requires an additional proof for the dual form. The authors provide this proof, which defines the core of their theoretical contribution.

### Strengths
- The paper is well structured and clearly highlights contributions of the work.
- The results are convincing: the paper gives both theoretical insights and the corresponding practical implications with competitive empirical results on standard deep learning classification problems (MNIST, CIFAR10, CIFAR100; Table 1), and improved robustness scores for increasing adversarial attack strengths (Table 2).

### Weaknesses
- In contrast to what the authors write at the end of section 3 in "Theoretical contribution and comparison to previous work", my impression is that Sinha et al's approach is actually based on the theoretical foundation of Blanchet & Murthy 2019 (Sinja et al. cite the version that was put on arxiv in 2016).
- The adapted cost function for incorporating all three AT approaches is unbounded and is used as a smoothed version in practice to be differentiable (eq 16). This change makes sense, however, the authors do not elaborate on the effects this might have on the theoretical insights.
- A clear comparison to the WRM approach by Sinha et al. (2017) is missing. Given that this is a related approach that already connects WDR with AT (limited to PGD-AT), it is important to highlight their differences. The authors do describe the approach by Sinha et al. (2017) but do not highlight the exact theoretical difference to their approach. Additionally, they do not show empirical results for WRM on the example problems.
- Some of the empirical results appear a bit oversold, e.g. the authors state that in Table 1 their approach “boosts the model robustness” compared to the other AT methods. However, in many cases the difference seems relatively small, on the order of 0.1, or 1.0 percent. It is clear that the performance of their approach is competitive, but to make strong statements about them the results would need error bars, e.g., standard errors of the mean over 5 or 10 repetitions of the benchmark.
- A detailed discussion of related work happens only in the appendix which is not part of the core submission and cannot be taken into account for evaluating the submission. It would be good to move part of that discussion to the main paper.
- The paper has quite some grammatical errors like misplaced commas, missing words misspellings, which impedes clarity and readability.

### Suggestions for improvements
- If possible, find an example used in Sinha et al. (2017) to compare to, e.g., their MNIST example. If their code is not available or difficult to tune, it should be possible to conduct the experiment with your methods and compare with the numbers in their paper.
- I do not think that SOTA is necessary for your paper to get accepted. However, I think that this is an important topic for the DNN community and that it should therefore be clearly communicated which approaches work well in which scenarios, e.g., does WRM work better for PGD-AT attacks?
- weakening of the statements about the results in Table 1 and 2 and to conduct additional repetitions of the benchmarks to have error bars on the numbers in the table.
- Add an actual figure caption for figure 1.
- Clarify the theoretical contributions of your approach vs. Sinha et al. (2017), e.g., that their work is based on Blancet & Murthy (2019) as well.
- Some references are not correct, e.g., Sinha et al. 2017 is not a preprint, it was actually accepted as an oral contribution to ICML 2018. Please check all your references and make sure you credit previous work correctly.

### Questions:
- Why is WRM fundamentally the same as PGD with lambda fixed? In WRM lambda is fixed as a penalty parameter before optimization, but this does not mean that it is the same as PGD. Could you clarify this statement?

### Typos/minor points
- There are several occasions in the paper where the definitive article to a noun, or the plural to make it indefinite, is missing e.g., already in the second sentence of the abstract.
- paragraph between equations 2 and 3: iif <- iff
- paragraph between equations 3 and 4: relaxation <- relaxed
- section 2.2 first paragraph: was AML defined before?
- sentence between equations 8 and 9: “one can derive to Eq. (9)” maybe you mean “one can arrive at” or “derive Eq (9)”?
- paragraph after equation 10: “The see why the later is the case” <- “To see why the latter is the case”
- paragraph after equation 12: “upon on” <- “on” (?)
- paragraph after equation 15: “theory developed in (B & M., 2019)” <- “the theory developed in B & M (2019).
- There are other occasions where the “in text” vs “in parentheses” citations are mixed up, please double check.
- section 4 second sentence: “we first discuss about” <- “we first discuss”

and it is likely that I missed some, so please make sure to check for typos carefully.


**Summary Of The Paper:**

The paper proposes a new approach for improving adversarial training (AT) in deep neural networks (DNN). While previous approaches generated point-wise adversary examples to enrich the training data set, this approach extends previous work on Wasserstein distributional robustness (WDR) and suggests a unified WDR framework to encompass previous approaches and to improve AT. The theoretical results show that previous approaches can indeed be formulated in the WDR framework. The empirical results show that the newly derived methods for AT improve the robustness to adversarial attacks on several classic DNN example problems.

**Summary Of The Review:**

The paper makes a theoretical contribution by extending the work on WDR for AT to other AT approaches. It shows promising results of their unifying framework on common example problems. Therefore, taking into account the weaknesses listed above I vote for a score marginally above the acceptance threshold. However, I am willing to increase my score if the weaknesses listed above are addressed, especially a thorough comparison to the WRM approach by Sinha et al (2017) (~500 citations, oral paper at ICML 2018).

---

> ### Author Response · Authors · 2021-11-19
> **Response to Reviewer 1ivF (part 1)**
>
> We thank the Reviewer for your positive comments. Please find our response as below:
>
> **1. In contrast to what the authors write at the end of section 3 in
> "Theoretical contribution and comparison to previous work", my impression is that
> Sinha et al's approach is actually based on the theoretical foundation of Blanchet \& Murthy
> 2019 (Sinja et al. cite the version that was put on arxiv in 2016).**
>
> We agree that there are close connections between Sinha et al.  (2017) and Blanchet \& Murthy (2019). But due to some differences, it might be a bit too strong to state that the former relies on the latter.
> We provided a detailed discussion in Section 2.1 in the main paper. In general, the primal form of Sinha et al.  (2017) is the left side of Eq. (4), while its dual form is the right side of Eq. (4). Meanwhile, the primal form of Blanchet \& Murthy (2019) is Eq. (1), while its dual form is Eq. (3). The key difference of two works is that in Sinha et al.  (2017), $\lambda$ is a predefined hyper-parameter, while in Blanchet \& Murthy (2019), $\lambda$ is learnable.
>
> **2. The adapted cost function for incorporating all three AT approaches
> is unbounded and is used as a smoothed version in practice to be differentiable (eq 16).
> This change makes sense, however, the authors do not elaborate on the effects this might
> have on the theoretical insights.**
>
> Thanks for this suggestion. As the reviewer pointed out,  we did this adaptation mainly from a practical perspective. We agree that it is very interesting to provide theoretical study on how the tightness in approximation of the smooth cost function to the ideal one or how the convergence rate of the smooth cost function to the ideal one affects the smoothness.
>
> **3. A clear comparison to the WRM approach by Sinha et al. (2017) is missing.
> Given that this is a related approach that already connects WDR with AT (limited to PGD-AT),
> it is important to highlight their differences. The authors do describe the approach by
> Sinha et al. (2017) but do not highlight the exact theoretical difference to their approach.
> Additionally, they do not show empirical results for WRM on the example problems**
>
> The performance of WRM highly depends on the Lagrange dual parameter $\gamma$ (or $\epsilon=0.5/\gamma$ in their implementation in [1]), which controls the robustness level. As mentioned in their paper, with large $\gamma$, the method is less robust but more tractable. Generally, decreasing $\gamma$ will reduce the natural accuracy but increase the robustness of the model as shown in Table below.
>
> |                 | Nat  | PGD  | AA   | B&B  |
> |-----------------|------|------|------|------|
> | $\epsilon=0.1$  | 90.9 | 15.3 | 13.7 | 15.8 |
> | $\epsilon=0.5$  | 86.7 | 33.9 | 32.6 | 35.4 |
> | $\epsilon=1.0$  | 83.7 | 40.9 | 39.8 | 41.4 |
> | $\epsilon=2.0$  | 79.4 | 45.4 | 43.6 | 45.5 |
> | $\epsilon=5.0$  | 71.6 | 47.5 | 45.2 | 46.2 |
> | $\epsilon=10.0$ | 65.0 | 46.6 | 43.4 | 44.4 |
>
> We obtained the best performance on MNIST with $\gamma=0.05$ (CNN), while on CIFAR10 and CIFAR100 with $\gamma=0.5$ (ResNet18). The best results with three benchmark datasets have been reported as in table below. It is a worth mentioning that while we could obtain a similar performance as reported Sinha et al. (2017) on the MNIST dataset with their architecture (3 Convolution layers + 1 FC layer), however, WRM seems much less effective with larger architectures.
>
> |         | MNIST |      |      |      | CIFAR10 |      |      |      | CIFAR100 |      |      |      |
> |---------|:-----:|------|------|------|:-------:|------|------|------|:--------:|------|------|------|
> |         | Nat   | PGD  | AA   | B&B  | Nat     | PGD  | AA   | B&B  | Nat      | PGD  | AA   | B&B  |
> | WRM     | 91.8  | 27.1 | 4.5  | 8.2  | 83.7    | 40.9 | 39.8 | 41.4 | 56.6     | 24.7 | 21.3 | 22.9 |
> | PGD-AT  | 99.4  | 94.0 | 88.9 | 91.3 | 86.4    | 46.0 | 42.5 | 44.2 | 72.4     | 41.7 | 39.3 | 39.6 |
> | UDR-PGD | 99.5  | 94.3 | 90.0 | 91.4 | 86.4    | 48.9 | 44.8 | 46.0 | 73.5     | 45.1 | 41.9 | 42.3 |
>
> [1] https://github.com/duchi-lab/certifiable-distributional-robustness/blob/master/attacks\_tf.py

---

> > ### Author Response · Authors · 2021-11-19
> > **Response to Reviewer 1ivF (part 2)**
> >
> > **4. Some of the empirical results appear a bit oversold, e.g. the authors
> > state that in Table 1 their approach “boosts the model robustness” compared to the other
> > AT methods. However, in many cases the difference seems relatively small, on the order
> > of 0.1, or 1.0 percent. It is clear that the performance of their approach is competitive,
> > but to make strong statements about them the results would need error bars, e.g.,
> > standard errors of the mean over 5 or 10 repetitions of the benchmark**
> >
> > As reported in Table 2, the average improvement of our UDR-PGD over PGD-AT is more than $3\%$, while that for UDR-MART over MART is also around $3\%$.
> > While our methods achieve competitive performance compared to methods in Robustbench as shown in Table 4, the additional experiments on C\&W attack as shown in Table below demonstrate that our method significantly improve the standard AT methods by a large margin (around 5\% over PGD-AT and AWP-AT and 2\% over TRADES, detail can be found in Appendix F). Therefore, we would like to keep our argument.
> >
> > |            |  Nat  | c=0.5 | c=0.7 | c=1.0 | Avg-Gap |
> > |------------|:-----:|:-----:|:-----:|:-----:|:-------:|
> > | PGD-AT*    | 84.93 | 40.85 | 25.90 | 12.95 |    -    |
> > | UDR-PGD*   | 84.60 | 47.31 | 31.58 | 16.57 |   5.25  |
> > | TRADES     | 85.70 | 47.65 | 34.30 | 21.03 |    -    |
> > | UDR-TRADES | 84.93 | 49.14 | 36.33 | 23.28 |   1.92  |
> > | AWP-AT     | 85.57 | 49.91 | 34.31 | 18.97 |    -    |
> > | UDR-AWP-AT | 85.51 | 54.44 | 39.86 | 23.61 |   4.91  |
> >
> > Because the performance is quite stable after several repetitions (with variance around $\pm 0.1\%$), therefore, we choose to report results with a single run as similar as in literature (e.g., [2]).
> >
> > [1] https://github.com/Harry24k/adversarial-attacks-pytorch
> >
> > [2] Pang et al., ICLR 2020. Bag of Tricks for Adversarial Training.
> >
> > **5. Why is WRM fundamentally the same as PGD with lambda fixed? In WRM lambda is fixed as a penalty parameter before optimization, but this does not mean that it is the same as PGD. Could you clarify this statement?**
> >
> > WRM is not equivalent to PGD when $\lambda$ is fixed because the closeness of WRM adversarial examples to their benign ones is controlled by $\lambda$. However, they are similar in the sense that these methods can only utilize local information of relevant benign examples when crafting adversarial examples.
> >
> > **6. Comments on the writing and Suggestions for improvements**
> >
> > Thanks for comments and we have updated the paper based on your suggestions.

---

> > > ### Comment · Reviewer_1ivF · 2021-11-29
> > > **Response to the authors comments on the reviews**
> > >
> > > Thank you for the additional explanations and the additional results.
> > >
> > > I maintain that the discussion of the previous work by Sinha et al. 2018 is not sufficient in some parts of the paper. E.g., the authors still write
> > > > Different from WRM (Sinha et al., 2018) , our proposed framework is developed based on theoretical foundation of (Blanchet & Murthy, 2019).
> > >
> > > However, Sinha et al. 2018 is indeed relying on Blanchet & Murthy 2019 (there are some differences of course, e.g., the dependence on lambda), while this sentence suggests something else. Thus, I ask the authors to clarify the dependence and change the sentence accordingly, to avoid confusion by the reader and to acknowledge previous work appropriately.
> > >
> > > Regarding the wording for describing the empirical results: I agree that the results stated in Table 2 are convincing. However, I was referring to the results presented in Table 1, and the wording used to describe them, e.g., ``boost the model performance significantly’’. This seems inadequate given that the standard deviation over repetitions is 0.1 and the number are so close to each other.
> > >
> > > Lastly, I want to point out that the second sentence in the abstract is still off grammatically, e.g., you could make AT plural to fix that. Additionally, I find it difficult to parse and would recommend restructuring it.
> > >
> > > Overall, I will leave my evaluation unchanged.

---

> > > > ### Author Response · Authors · 2021-11-30
> > > > **Response to Reviewer 1ivF**
> > > >
> > > > We thank the Reviewer for your effort to review our paper and your helpful comments. We will revise the abstract and soften the discussion regarding Table 1 as your suggestion in the revision.

---

### Official Review · Reviewer_7bWw · 2021-11-02

**Correctness:** 3
**Technical Novelty And Significance:** 3
**Empirical Novelty And Significance:** 3
**Recommendation:** 8
**Confidence:** 2

**Main Review:**

The paper is overall well-written and interesting to read. The proposed method, while unifying previous methods, performs better than previous methods (PGD, TRADES and MART). The proofs are not carefully checked.

My major comments follow.
1.	It is proved that (14) is equivalent to the standard adversarial training objective, but is (14) the dual program of the Wasserstein distributional robust optimization? It seems that the equivalence of the dual program requires certain constraints on the function $g_\theta$.
2.	I understand that why you need to use sign of the gradient of $\hat{c_{\mathcal{X}}}$ in practice. But I feel the need of more explanations to show where the difficulty comes from. For example, if the objective function (17) is fundamentally good for adversarial training, then it may not need such tricks. More concisely, I wonder if it is just an empirical trick or there are more underlying reasons.
3.	I’d like to understand more about the reasons of the empirical success of the proposed method. I see that the authors contribute some of the success to “soft-ball projection” and “utilizing the global information”, but I’m curious to understand more beyond the general discussion, e.g., theoretical analysis about the empirical success. I understand that in-depth theoretical analysis may worth another paper, but it would be great if the authors can provide more insights.


**Summary Of The Paper:**

This paper proposes a unified framework for adversarial training from the perspective of Wasserstein distributional robustness. It is proved that some standard previous adversarial training methods are special cases of the proposed framework. The paper also proposes an adversarial training method given the framework. Empirical results demonstrate that the proposed method is more effective than previous methods.

**Summary Of The Review:**

The proposed method is novel and interesting to me. The empirical results seem promising. Overall, I think the paper has a decent empirical contribution, and it also has the potential to offer a unified theoretical explanation for adversarial training. Unless I misunderstand some key parts of the paper, I recommend accepting.

---

> ### Author Response · Authors · 2021-11-19
> **Response to Reviewer 7bWw (part 1)**
>
> We thank the Reviewer for your positive comments. Please find our response as below:
>
> **1. It is proved that (14) is equivalent to the standard adversarial
> training objective, but is (14) the dual program of the Wasserstein distributional
> robust optimization? It seems that the equivalence of the dual program requires
> certain constraints on the function $g_{\theta}$.**
>
> In our proof, the only constraint we need is that function $g_\theta$ is a upper semi-continuous function (please refer to Assumption 2 in  Blanchet \& Murthy (2019)). This constraint is trivial because $g_\theta$ is always a continuous function.
>
> **2. I understand that why you need to use sign of the gradient of $\tilde{c}_ {\mathcal{X}}$
> in practice. But I feel the need of more explanations to show where the difficulty
> comes from. For example, if the objective function (17) is fundamentally good
> for adversarial training, then it may not need such tricks. More concisely,
> I wonder if it is just an empirical trick or there are more underlying reasons.**
>
> Yes, the optimization in Eq. (18) is similar to other AT method and we can apply iterative projected gradient ascent to optimise it in principle, similarly as PGD. However, in practice, it has a numerical issue that cannot be solved with few iterations (e.g., 10 iterations on experiments with the CIFAR10 dataset).
>
> More specifically, if PGD is applied, at each update step (before projection), the adversary example moves a step $grad= \nabla_{x'} \left( g_ {\theta} (x',x,y) - \lambda \hat{c}_ {\mathcal{X}} \left(x',x\right) \right) = \nabla_{x'}g_ {\theta}(x',x,y) - \lambda \nabla_{x'}\hat{c}_ {\mathcal{X}}\left(x',x\right)$. However, as shown in Figure 7 in the Appendix, the strength of
> gradient $grad1=\nabla_{x'}g_ {\theta}(x',x,y)$ is much smaller than
> $grad2=\nabla_{x'}\hat{c}_ {\mathcal{X}}\left(x',x\right)$. For example,
> on the MNIST dataset, $grad1\in[-5\times10^{-4},5\times10^{-4}]$
> while $grad2\in[-0.3,0.3]$ which is 600 times larger.
> Therefore, if using single update step, the gradient $\nabla_{x'}\hat{c}_ {\mathcal{X}}\left(x',x\right)$
> dominates the other and pulls the adversarial examples close to the
> natural input.
> While a global optimal can be reached with sufficiently large iterative steps, however, with few iterations, it likely biases toward the $grad2$'s direction. Eventually, these adversarial examples are weaker and do not help improve the robustness. Alternatively, we break single update step
> for solving Eq. (18) to two sub-steps. Because we use $sign(grad1)$ instead of $grad1$ as in PGD in the first sub-step, therefore, we also use $sign(grad2)$ in the second sub-step to balance between push/pull steps.

---

> > ### Author Response · Authors · 2021-11-19
> > **Response to Reviewer 7bWw (part 2)**
> >
> > **3. I’d like to understand more about the reasons of the empirical
> > success of the proposed method. I see that the authors contribute some of the
> > success to “soft-ball projection” and “utilizing the global information”, but I’m
> > curious to understand more beyond the general discussion, e.g., theoretical analysis
> > about the empirical success. I understand that in-depth theoretical analysis may
> > worth another paper, but it would be great if the authors can provide more insights.**
> >
> > We are very happy to share our further thoughts about the reasons of the empirical success of our proposed method.
> >
> > Our key observation is that either standard adversarial training methods or Wasserstein distrbutional robustness methods get benefit from more diverge and geometry-aware adversarial examples [1].
> >
> > For WRM in Sinha et al.  (2017), the closeness of the adversarial examples and their benign ones is controlled by $\lambda$ (please refer to Eq. (4) in Section 2.1 of the main paper). However, tuning a single value $\lambda$ for all data examples seems implausible. We observe that  for WRM if we set $\lambda$ small, adversarial examples are too distant to their benign ones for which most of them stay distantly outside the $\epsilon$-balls $B_\epsilon$. In contrast, if we set $\lambda$ high, they become too close.
> >
> > For the standard approaches like PGD and TRADES, because of projecting onto the hard balls, we find that a large portion of adversarial examples lying on the balls. Moreover, for our versions, by using the concept of softball, we can generate more diverge and flexible adversarial examples, e.g., some lie inside the balls and some stay a bit outside the balls. These intuition and motivation are confirmed by the fact that our proposed approaches really outperform the baselines for defending attacks under larger $\epsilon$ (please refer to the experimental results in Table 2).
> >
> > [1] Zhang et al., ICLR 2021. Geometry-Aware Instance-Reweighted Adversarial Training.

---

### Official Review · Reviewer_wEoc · 2021-11-03

**Correctness:** 4
**Technical Novelty And Significance:** 2
**Empirical Novelty And Significance:** 2
**Recommendation:** 5
**Confidence:** 3

**Main Review:**

Strength:
Author(s) seem to be aware of the related literature based on their references to prior works, and also the exact description of the said works. Mathematical statements are rigor and correct as far as I have checked. Moreover, paper is very well-written.

Weaknesses:
Even though paper explicitly insists that its proposed theoretical results are $\underline{\mathrm{not}}$ trivial, at least I, as a reader, have failed to understand their significance. The relation between point-wise adversaries which are commonly used in AT-like methods and W-DRO is already known to the community. This fact has been mentioned even by the author(s) themselves; For example, see their reference to Staib and Jegelka (2017), or the seminal work of Sinha et al. (2017). I rely on author(s)' claim that results are new, since no prior work has tried to derive explicit relations similar to Theorems 1 and 2 of the current paper. Still, the current results do not seem surprising and worthy of publication at ICLR.

Additional note: paper claims that results are not trivial since the dual form that is usually utilized in W-DRO assumes a bounded loss function, while this condition has been relaxed here. TBH, I haven't checked the proofs, completely. However, no sophisticated mathematical tools have been used throughout the paper and proofs are rather simple and short. It would be more informative if author(s) can give more insight about the technical difficulties that they have faced while deriving the results and thus shed light on their possible significance.

Paper also claims some experimental contributions. I am not completely familiar with the experimental side of this line of research, so I wait to see other reviewers' comments on that matter.

-------------------------------

Minor comments and suggestions:

"The see why" -> "To see why"

**Summary Of The Paper:**

This paper aims to establish a formal relation between Wasserstein distributionally robust optimization (W-DRO) and a number of Adversarial Training (AT)-like defenses against adversarial attacks. In particular, three well-known defenses, namely PGD-AT, TRADES and MART have been the center of attention.

Paper derives a theory that shows each of the above methods are special cases of a W-DRO, settled in a properly-defined augmented space.

**Summary Of The Review:**

Paper seems to lack enough theoretical novelty at this stage. My current vote is Weak Reject, but I also like to see other comments and feedbacks.

---

> ### Author Response · Authors · 2021-11-19
> **Response to Reviewer wEoc (part 1)**
>
> We thank the Reviewer for your feedback. Please find our response to your remain concerns as below:
>
> **1. Even though paper explicitly insists that its proposed theoretical results are not
> trivial, at least I, as a reader, have failed to understand their significance. The relation between
> point-wise adversaries which are commonly used in AT-like methods and W-DRO is already known to
> the community.**
>
> Firstly, from the theoretical perspective,
> our method extends the the theoretical framework in Blanchet \& Murthy (2019) into a WS-based distributional robustness defense method.
>
> Our theoretical development is highly non-trivial.
> Specifically, it requires an elegant technique to bypass a restrictive assumption of the finiteness of the cost function $\tilde{c}_{\mathcal{X}}(x,x')$ so that the duality holds.
>
> The main obstacle comes from the fact that $\tilde{c}_{\mathcal{X}}(x,x')$ is not totally finite.
> Comparing with Sinha et al. (2017), ours is principally different in the adaptive capability of $\lambda$, which is always fixed in Sinha et al. (2017). This difference opens a door for our softball projection to flexibly control the distances between adversarial and benign examples.
>
> Secondly, from the practical perspective, although Sinha et al. (2017) can be viewed as a novel work in leveraging distributional robustness to adversarial defense problems, to our best of knowledge, there was not evident sign presented in that paper showing their proposed approach works for a more complex dataset like CIFAR10 and CIFAR100. By using the tool of softball projection and the adaptive capability of $\lambda$ hinted from our developed theory, we demonstrate distributional approaches can obtain satisfactory performance on real-world datasets, compared with many SOTA methods. Furthermore, the additional experiment with C\&W attack on WideResNet in table below shows that our methods significantly improve the robustness over standard AT methods with a large margin. Therefore, we believe that the experimental results are comprehensive and convincing to demonstrate the significance of our method.
>
> |            |  Nat  | c=0.5 | c=0.7 | c=1.0 | Avg-Gap |
> |------------|:-----:|:-----:|:-----:|:-----:|:-------:|
> | PGD-AT*    | 84.93 | 40.85 | 25.90 | 12.95 |    -    |
> | UDR-PGD*   | 84.60 | 47.31 | 31.58 | 16.57 |   5.25  |
> | TRADES     | 85.70 | 47.65 | 34.30 | 21.03 |    -    |
> | UDR-TRADES | 84.93 | 49.14 | 36.33 | 23.28 |   1.92  |
> | AWP-AT     | 85.57 | 49.91 | 34.31 | 18.97 |    -    |
> | UDR-AWP-AT | 85.51 | 54.44 | 39.86 | 23.61 |   4.91  |

---

> > ### Author Response · Authors · 2021-11-19
> > **Response to Reviewer wEoc (part 2)**
> >
> > **2. paper claims that results are not trivial since the dual form that is usually
> > utilized in W-DRO assumes a bounded loss function, while this condition has been relaxed here.
> > TBH, I haven't checked the proofs, completely. However, no sophisticated mathematical tools have
> > been used throughout the paper and proofs are rather simple and short. It would be more
> > informative if author(s) can give more insight about the technical difficulties that they have
> > faced while deriving the results and thus shed light on their possible significance.**
> >
> > One of the most technical challenges we need to address in our work is that in the theory developed in Blanchet \& Murthy (2019), to equivalently transform the primal form to the dual form, it requires the cost function to be finite. However, the cost function $\tilde{c}_{\mathcal{X}}(x,x')$ is infinite if $x'$ falls outside of the ball $B(x,\epsilon)$. We need to provide proof of the equivalence of the primal and dual forms in our context.
> > Although the proof does not require sophisticated mathematical tools, it is underdone before and makes our work theoretically solid.
> >
> > In addition, from the practical perspective, the domination of the gradient of the cost function
> > $grad2=\nabla_ {x'}\hat{c}_ {\mathcal{X}}\left(x',x\right)$
> > over the gradient of the risk function $grad1=\nabla_ {x'}g_ {\theta}(x',x,y)$
> > in Eq. (18) is also an issue that prevents to reach optimal solution of the optimization in Eq. (18). More specifically, if applying PGD to optimize Eq. (18), at each update step (before projection), the adversary example moves a step
> > $grad= \nabla_{x'} (g_ {\theta}(x',x,y) - \lambda \hat{c}_ {\mathcal{X}}(x',x)) = \nabla_{x'}g_ {\theta}(x',x,y) - \lambda \nabla_{x'}\hat{c}_ {\mathcal{X}}(x',x)$.
> >
> > However, as shown in Figure 7 in the Appendix, the strength of
> > gradient $grad1=\nabla_{x'}g_ {\theta}(x',x,y)$ is much smaller than
> > $grad2=\nabla_{x'}\hat{c}_ {\mathcal{X}}\left(x',x\right)$. For example,
> > on the MNIST dataset, $grad1\in[-5\times10^{-4},5\times10^{-4}]$
> > while $grad2\in[-0.3,0.3]$ which is 600 times larger.
> > Therefore, if using single update step, the gradient $\nabla_{x'}\hat{c}_ {\mathcal{X}}\left(x',x\right)$
> > dominates the other and pulls the adversarial examples close to the
> > natural input. Because WRM used this approach, therefore, it could not learn a good adversarial examples to improve the robustness. Alternatively, to overcome this issue, we break single update step
> > for solving Eq. (18) to two sub-steps as introduce in Algorithm 1. The two sub-steps approach help us balance between push movement (toward direction of $grad1$ to maximize the risk function) and pull movement (toward direction of $grad2$ to minimize the cost function).
> >
> > **3. Comments on the writing**
> >
> > Thanks, we have updated the paper based on your suggestions.

---

### Official Review · Reviewer_T7rh · 2021-11-04

**Correctness:** 3
**Technical Novelty And Significance:** 3
**Empirical Novelty And Significance:** 3
**Recommendation:** 8
**Confidence:** 4

**Main Review:**

This is an excellently well-written paper that applies the idea of distributional robustness for crafting adversarial examples. The methodology in the paper extends that of (Sinha et. al. 2017) by introducing learnable parameters. Moreover, a soft-ball based projected gradient descent is used instead of a hard one. Extensive experiments are conducted to demonstrate that the proposed method UDR when applied in conjunction with existing perturbation methods, such as PGD and TRADES, produces better results on standard datasets.

One weakness is that the paper develops a white-box attack methodology, where the adversary (Algorithm 1) is assumed to have a complete knowledge of the model parameters (model parameters being simultaneously updated with the attack generation parameters). This makes the attack algorithm not that realistic.

Some specific comments:

1. It's a bit strange to see the symbol \infty in an algebraic equation (Equation 10). The usual way of writing this would be with a symbol M, and stating that M is larger than M_0.
2. The authors should explain precisely what is meant by "lower semi-continuous", and argue why that is the case with the defined cost function.
3. CE (the cross-entropy loss) is currently undefined. It should be defined in a similar manner as the definition of BCE.
4. stay closely --- stay close
5. For the blackbox attacks (reported in Page 8), do you also learn the model parameters by Algorithm 1?





**Summary Of The Paper:**

This paper applies the idea of distributional robustness for crafting adversarial examples. The methodology in the paper extends that of (Sinha et. al. 2017) by introducing learnable parameters. Moreover, a soft-ball based projected gradient descent is used instead of a hard one.


**Summary Of The Review:**

I'm leaning towards an accept due to the use of the novel idea of soft-ball projection and the demonstration that the proposed method works on a number of different standard datasets.

---

> ### Author Response · Authors · 2021-11-19
> **Response to Reviewer T7rh**
>
> We thank the Reviewer for your positive comments. Please find our response to your remain concerns as below:
>
> **1. One weakness is that the paper develops a white-box attack methodology, where the adversary (Algorithm 1) is assumed to have a complete knowledge of the model parameters (model parameters being simultaneously updated with the attack generation parameters). This makes the attack algorithm not that realistic.**
>
> In this paper, we mainly focus on improving the deep learning robustness in
> the white-box setting, which is the most active research area in adversarial machine learning.
> In addition, we have explored our model's robustness in the black-box setting in the case of transferring attacks as reported in Table 3.
> We agree that more study is needed in the black-box setting, which will be our future work. Thanks for your suggestion.
>
> **2. For the black-box attacks (reported in Page 8), do you also learn the model parameters by Algorithm 1?**
>
> In Table 3 (page 8) we reported the robustness evaluation in black-box attack setting, i.e., adversarial examples have been crafted by an adversary w.r.t. a **Source** model and be transferred to attack **Target** models. In this experiment, we did not choose our Algorithm 1 but PGD attack to craft adversarial examples which is a common black-box setting.
>
> **3. The authors should explain precisely what is meant by "lower semi-continuous", and argue why that is the case with the defined cost function.**
>
> We have revised the paper to make it clearer as below (showing in blue in the updated paper, page 4):
>
> "where we note that this cost function is non negative, satisfies $c(z,z)=0$ and lower semi-continuous, i.e., $ \underset{z' \rightarrow z_0}{\lim} \inf c(z,z') \geq c(z,z_0) $."
>
> Moreover, the lower-semi continuous property of the cost function is essential for optimal transport theory in general and the theoretical results in  Blanchet \& Murthy (2019) and Sinha et al. (2017) in specific due to some following theoretical properties that are useful for proofs.
>
> - Given a lower-semi continuous function, there exists a sequence of increasing continuous functions (or more strongly functions satisfying Lipschitz condition) point-wisely converging to this function.
>
> - There exists a global minimum for a lower-semi continuous function on a compact set.
>
> Last but not least, most popular cost function in machine learning is continuous, hence being lower-semi continuous is a reasonable assumption.
>
> **4. Comments on the writing**
>
> Thanks and we have updated the paper based on your suggestions.

---

### Author Response · Authors · 2021-11-19
**Response Hightlights**

We thank all reviewers for their time and efforts in reviewing the paper and providing helpful suggestions/comments. The major changes have been highlighted in blue in the revision.
In what follows, we highlighted our contributions, especially, comparing with the previous work of Sinha et al. (2017).

**Contributions to previous work**. In this paper, we believe that we have two considerable contributions from both theoretical and practical perspectives.

First, from the theoretical perspective, based on the theoretical framework in Blanchet \& Murthy (2019) serving as a general framework for Wasserstein (WS) distributional robustness, we develop theoretical-grounded approach for a WS-based distributional robustness defense method. Moreover, our theoretical development is highly non-trivial if you observe our proofs in the appendix for which we need to offer an elegant technique to bypass a restrictive assumption of the finiteness of the cost function $ \tilde{c}_{\mathcal{X}} (x, x') $ so that the duality holds.

The main obstacle comes from the fact that $\tilde{c}_{\mathcal{X}}(x, x')$ is not totally finite. Comparing with Sinha et al. (2017), ours is principally different in the adaptive capability of $\lambda$, which is always fixed in Sinha et al. (2017). More specifically, the adaptive capability of opens a door for our softball projection to flexibly control the distances between adversarial and benign examples.

Second, from the practical perspective, although Sinha et al. (2017) can be viewed as a seminal work in leveraging distributional robustness to defense problem, to our best of knowledge, there was not evident sign presented in that paper showing this approach worked for a more complex dataset like CIFAR10 and CIFAR100. By using the tool of softball projection and the adaptive capability of $\lambda$ hinted from our developed theory, we demonstrate distributional approaches can obtain satisfactory performance on real-world datasets. Furthermore, the additional experiment with C\&W attack on WideResNet in table below shows that our methods significantly improve the robustness over standard AT methods with a large margin (the detail discussion has been added to Appendix F). Therefore, we believe that the experimental results are comprehensive and convincing to demonstrate the significance of our method.

|            |  Nat  | c=0.5 | c=0.7 | c=1.0 | Avg-Gap |
|------------|:-----:|:-----:|:-----:|:-----:|:-------:|
| PGD-AT*    | 84.93 | 40.85 | 25.90 | 12.95 |    -    |
| UDR-PGD*   | 84.60 | 47.31 | 31.58 | 16.57 |   5.25  |
| TRADES     | 85.70 | 47.65 | 34.30 | 21.03 |    -    |
| UDR-TRADES | 84.93 | 49.14 | 36.33 | 23.28 |   1.92  |
| AWP-AT     | 85.57 | 49.91 | 34.31 | 18.97 |    -    |
| UDR-AWP-AT | 85.51 | 54.44 | 39.86 | 23.61 |   4.91  |

---

### Decision · Program_Chairs · 2022-01-20

**Decision:**

Accept (Poster)

**Comment:**

The paper extends the previously established connection between adversarial training (AT) and Wasserstein distributional robustness (WDR) to other adversarial defense methods such as PGD-AT, TRADES and MART, and connects them to WDR. While this connection itself is not surprising given earlier works connecting AT and WDR, the paper makes contributions in establishing it formally and proposing algorithmic variations (eg, softball projection) that show clear empirical gains on standard benchmarks of MNIST/CIFAR10/CIFAR100 over point-wise adversarial defense methods.